# Improved Water Vapour retrieval from AMSU-B/MHS in the Arctic

Arantxa M. Triana-Gómez[1], Georg Heygster[1], Christian Melsheimer[1], Gunnar Spreen[1],
Monia Negusini[2], and Boyan H. Petkov[3]

[1]Institute of Environmental Physics, University of Bremen, Bremen, Germany
[2]Institute of Radio Astronomy, INAF, Bologna, Italy
[3]Institute of Atmospheric Sciences and Climate, CNR, Bologna, Italy

**Correspondence:** Arantxa M. Triana-Gómez (aratri@uni-bremen.de)

**Abstract.**

Monitoring of water vapour in the Arctic on long time scales is essential for predicting Arctic weather and understanding climate trends, as well as addressing its influence in the positive feedback loop contributing to Arctic Amplification. However, this is challenged by the sparseness of in-situ measurements and the problems that standard remote-sensing retrieval methods for water vapour have in Arctic conditions. Here, we present advances in a retrieval algorithm for vertically integrated water vapour (total water vapour, TWV) in polar regions from data of satellite-based microwave humidity sounders: (1) In addition to AMSU-B (Advanced Microwave Sounding Unit-B), we can now also use data from the successor instrument MHS (Microwave Humidity Sounder); (2) artefacts caused by high cloud ice content in convective clouds are filtered out. Comparison to in-situ measurements using GPS and radiosondes during 2008 and 2009 as well as to radiosondes during the N-ICE2015 campaign and to ERA5 reanalysis show overall good performance of the updated algorithm.

## 1 Introduction

Water vapour is a key element of the hydrological cycle (Chahine, 1992; Serreze et al., 2006; Jones et al., 2007; Hanesiak et al., 2010), with shifts in it affecting atmospheric transport processes, creating and intensifying droughts and flooding (Trenberth et al., 2013). Additionally, as the most important greenhouse gas in the atmosphere, it has a dominant effect on climate and radiative forcing (Soden et al., 2002; Dessler et al., 2008; Kiehl and Trenberth, 1997; Trenberth et al., 2007; Ruckstuhl et al., 2007). Hence, it is essential to monitor its variability considering both that water vapour increases when temperature does and the anthropogenic increase of other greenhouse gases (Solomon et al., 2010), with the water vapour positive feedback loop highlighted as part of other feedbacks responsible for Arctic Amplification (Francis and Hunter, 2007; Miller et al., 2007; Screen and Simmonds, 2010; Ghatak and Miller, 2013). In summary, understanding the water vapour cycle has high value, yet our comprehension is incomplete (Stevens and Bony, 2013). Throughout this paper, when mentioning atmospheric water content, we refer to the vertically integrated mass in an air column with an area of 1 m$^2$, and call it total water vapour (TWV, sometimes also called column water vapour, integrated water vapour or total precipitable water), the units are hence kg/m$^2$.

Balloon-borne radiosondes are a standard method for retrieving the water vapour profile. Additionally, ground-based retrievals by microwave radiometers as well as GPS-based retrievals – while having a lower vertical resolution – are good for

monitoring purposes in regions where ground stations can be installed. However, in the Arctic, neither radiosondes measurements nor ground-based retrievals are sufficient for this purpose because weather stations are too scarce. Only satellite measurements fulfill the global coverage requirements. An additional challenge is to construct a consistent long-term climate record, due to the changes in measuring instruments, and degradation of the existing ones. Because of the strong absorption properties of water vapour in the infrared and microwave range, suitable space-borne instruments can in principle ensure a complete global coverage of water vapour retrievals (Miao et al., 2001; Bobylev et al., 2010). In polar regions, however, satellite retrieval of water vapour faces a number of obstacles such as cloud cover which restricts infrared measurements, or incomplete understanding of the high and highly variable sea-ice emissivity which challenges microwave measurements. Some studies – like the one by Weaver et al. (2017) – have been done for TWV in the Arctic atmosphere, but none of them have been able to provide a long-term Arctic-wide data set.

An important step for Arctic water vapour retrieval comes from the work of Miao et al. (2001). They used data from the SSM/T2 (Special Sensor Microwave Humidity) humidity sounder to develop an algorithm which was designed to work in the Antarctic. The key concept of this method is the use of several microwave channels with similar surface emissivity but different water vapour absorption. These are the three channels near the 183.31 GHz water absorption line (183.31 $\pm 1$, $\pm 3$ and $\pm 7$ GHz), which, together with the channel at the 150 GHz window frequency, allows retrieval of TWV values up to about 7 kg/m$^2$. Above this value, two of the 183.31 GHz band channels become saturated and the sensor is not able to "see" through the whole atmospheric column anymore. In other words, when the TWV reaches a certain threshold, the brightness temperature at these AMSU-B channels does not change with increasing TWV (Miao, 1998; Melsheimer and Heygster, 2008). This limited range is enough for Antarctica, and suffices for the Arctic in winter conditions (in the polar winter atmosphere, the water vapour column is typically around 3 kg/m$^2$ according to Serreze et al. (1995)), as well as for the central Arctic (above 70° N) most of the year. However, because of the upper limit, this method cannot ensure monitoring of the complete yearly cycle. The algorithm developed by Melsheimer and Heygster (2008) extends the TWV retrieval range over sea ice by including the AMSU-B (Advanced Microwave Sounding Unit-B) 89 GHz channel into the retrieval. Using the triplet of the 183.31$\pm$7, 150 and the 89 GHz channels allows the retrieval to function up the saturation limit of the 183.31$\pm$7 GHz channel. This method has been compared with other datasets: In Rinke et al. (2009) a comparison with the HIRHAM model showed realistic patterns and maximum root-mean-square differences for monthly data in summer of 1-2.5 kg/m$^2$. For the comparison with Ny Ålesund radiosondes in Palm et al. (2010), the correlation coefficient was 0.86 and the slope 0.8$\pm$0.04. And lastly, in Buehler et al. (2012) AMSU-B TWV are compared to GPS data from Kiruna, with RMSD of 1 kg/m$^2$ and a correlation coefficient of 0.86. However, the AMSU-B algorithm is not without problem: while the frequency range allows it to bypass most clouds, the AMSU-B sensor is still sensitive to convective clouds with high ice content. Here we provide an approach for filtering out problematic data caused by the effect of such ice clouds. This is intended as groundwork for the planned merging with TWV retrieved over open ocean based on passive microwave imagers (product described by Wentz and Meissner, 2006).

In Section 2, we describe the algorithm in a more detailed way. In Section 3 we evaluate the application of the algorithm to MHS (Microwave Humidity Sounder) instead of AMSU-B data, which is necessary for extending the data set to cover recent years, performing a comparison with different in-situ data sources in Section 3.2 and to ERA5 reanalysis in Section 3.3. Then,

in Section 4 we evaluate the new ice cloud filtering developed for the algorithm in Section 4, and finally give some conclusions in Section 5.

## 2 Retrieval algorithm

### 2.1 Data sources

The algorithm uses microwave radiometer satellite measurements from humidity sounders such as AMSU-B or MHS on board the NOAA (National Oceanic and Atmospheric Administration) 15 to 19 satellites and EUMETSAT (European Organisation for the Exploitation of Meteorological Satellites) Metop-A, Metop-B and Metop-C satellites. The characteristics of each sensor can be found in Table 1, and the launch dates of each satellite in Table 2. Through this paper, when we refer to AMSU-B TWV, the brightness temperature data used for the retrieval is always from the sensor on NOAA-17, with the version from

the Fundamental Climate Data Record (Ferraro and Meng, 2016), which provides an inter-satellite calibrated set of brightness temperatures as described in Ferraro (2016). When we refer to MHS TWV, the brightness temperature data are from NOAA-18, similarly sourced.

    Additionally, to distinguish between surface types, the daily ice concentration provided by the ASI-algorithm (ARTIST Sea Ice algorithm, Spreen et al., 2008) is used, with pixels with ice concentrations below 15% as open water, while the ones with

more than 80% will be considered ice. The percentages between those will not be used.

### 2.2 Radiative transfer equation

The algorithm starts from the formulation of the radiative transfer equation in the contracted form by Guissard and Sobieski (1994) which describes the brightness temperature ($T_B$) measured by a space-borne radiometer as:

$$T_B(\theta) = m_p T_s - (T_0 - T_c)(1 - \epsilon_s)\mathrm{e}^{-2\tau \sec\theta}, \tag{1}$$

where $\theta$ is the zenith angle, $T_s$ and $T_0$ are the surface and air temperatures, respectively, $T_c$ is the cosmic background emission, $\epsilon_s$ the surface emissivity, $\tau_0$ the total opacity of the atmosphere in the vertical direction, and $m_p$ a correction to take into account both a non-isothermal atmosphere and the difference between the surface (skin) temperature, $T_s$, and the temperature of the atmosphere at the ground, $T_0$ ($m_p = 1$ would be the isothermal case and $T_0 = T_s$). The approach by Melsheimer and Heygster (2008), summarized in the following, assumes the ground to be approximated as a specular reflector, which should

be good enough for remote sensing in the frequency range we are dealing with, according to Hewison and English (1999).

### 2.3 Retrieval for equal emissivity assumption

Note that the entire derivation of the final total water vapour retrieval equation from the radiative transfer equation is described in detail in the initial paper for the Antarctic by Miao et al. (2001) and the subsequent Arctic extension by Melsheimer and Heygster (2008). We summarize it here because the basic mechanism is necessary to understand the changes performed.

We start from microwave radiometer satellite measurements in three different channels $i, j, k$, such as mentioned in Section 2.1. We assume none of these three channels are saturated, i.e., the sensor is still sensitive to the whole atmospheric column and ground. Additionally, we take the ground emissivity as equal in all three channels (as they see the same footprint, and the emissivity does not vary between the channels), while the water vapour absorption (mass absorption coefficient $k$ (m$^2$/kg)) is different, with $k_i < k_j < k_k$. Then, the brightness temperature difference of two channels $i,j$ can be expressed as:

$$\Delta T_{ij} \equiv T_{Bi} - T_{Bj} = (T_0 - T_c)(1 - \epsilon_s)(e^{-2\tau_i \sec\theta} - e^{-2\tau_j \sec\theta}) + b_{ij}, \tag{2}$$

where $\tau_i$ is the nadir opacity of the atmosphere at the frequency of channel $i$, and $b_{ij}$ is a bias related to the term $m_p$ for the channels $i$ and $j$:

$$b_{ij} = T_s(m_{pi} - m_{pj}), \tag{3}$$

As shown in Melsheimer and Heygster (2008) – Appendix II, the bias can here be approximated as:

$$b_{ij} \approx \int_0^\infty \left[ e^{-2\tau_i(z,\infty)\sec\theta} - e^{-2\tau_j(z,\infty)\sec\theta} \right] \frac{dT(z)}{dz} dz, \tag{4}$$

where $T(z)$ is the atmospheric temperature profile. Then we take the ratio of what we call compensated brightness temperature differences:

$$\eta_c \equiv \frac{\Delta T_{0ij}}{\Delta T_{0jk}} = \frac{\Delta T_{ij} - b_{ij}}{\Delta T_{jk} - b_{jk}} = \frac{e^{-2\tau_i \sec\theta} - e^{-2\tau_j \sec\theta}}{e^{-2\tau_j \sec\theta} - e^{-2\tau_k \sec\theta}}. \tag{5}$$

We can express the opacities $\tau_i$ as a sum of the atmospheric constituent contributions to them: water vapour ($\tau_i^w$) and oxygen ($\tau_i^{oxygen}$). The latter is negligible for AMSU-B channels near the water vapour line, so if we take water vapour mass absorption coefficients $k_i$ and TWV $W$:

$$\tau_i = \tau_i^w + \tau_i^{oxygen} \approx k_i W, \tag{6}$$

If we approximate the differences of exponentials by products in (5) and take logarithms, we get:

$$\ln(\eta_c) = B_0 + B_1 W \sec\theta + B_2 (W \sec\theta)^2 \tag{7}$$

The three constants $B_0$, $B_1$, and $B_2$ depend on the mass absorption coefficients for the different channels. The term quadratic in $W$ can be neglected (Selbach, 2003; Miao et al., 2001) which leaves us with an equation linear in $W$ that can then be solved to yield our retrieval equation:

$$W \sec\theta = C_0 + C_1 \ln(\eta_c) \tag{8}$$

where $C_0 = \frac{B_0}{B_1}$ and $C_1 = \frac{1}{B_1}$. They are determined empirically as calibration parameters from simulated brightness temperatures based on radiosonde profiles by a regression analysis, described in more detail below (Section 2.6).

## 2.4 Extension of the retrieval

Normally, for TWV values above 7 kg/m$^2$ , saturation occurs at Channel 19 (183.3±3 GHz). To extend the retrieval range above this threshold, another channel is required that is less sensitive to water vapour to take its place in the triplet. This means that a new set of assumptions has to be made about the surface emissivity influence. For AMSU-B, the next channel "in line"

is the one at 89 GHz (Channel 16). Thus, the three channels $i$, $j$, $k$ are now the AMSU-B Channels 16, 17 and 20 (89, 150 and 183.31±7 GHz). Because Channel 16 is so far from the other two, we can no longer assume that it has the same surface emissivity as the others. Therefore the retrieval equation needs to be re-derived with the changed premise: $\epsilon_i \neq \epsilon_j = \epsilon_k$. This leaves us with a similar looking retrieval equation:

$$W \sec\theta = C_0 + C_1 \ln(\eta_c') \tag{9}$$

where $\eta_c'$ is a modified ratio of compensated brightness temperatures:

$$\eta_c' \equiv \frac{r_j}{r_i}\left(\eta_c + C(\tau_j, \tau_k)\right) - C(\tau_j, \tau_k), \tag{10}$$

and $C(\tau_j, \tau_k)$ is defined as

$$C(\tau_j, \tau_k) = \frac{\mathrm{e}^{-2\tau_j \sec\theta}}{\mathrm{e}^{-2\tau_j \sec\theta} - \mathrm{e}^{-2\tau_k \sec\theta}}, \tag{11}$$

Since now there is a dependence on emissivities $\epsilon_i$, or, equivalently, on reflectivities $r_i = 1 - \epsilon_i$, the surface emissivity at

89 GHz needs to be examined. Ideally, the ratio of corresponding reflectivities would be taken for each footprint. However, that is not possible without knowing atmospheric conditions and surface temperature. As an approximation, the emissivity is parametrized, and fixed reflectivity ratios depending on surface types are obtained. This was done for sea ice in Melsheimer and Heygster (2008) and for open water surfaces in Scarlat et al. (2018). The upper limit of this extended retrieval is about 15 kg/m$^2$. Here, we will use this extended retrieval only over sea ice.

## 2.5 The "sub-algorithms": regime selection

As described through Sections 2.3 and 2.4, three different channel triplets are used for the retrieval, depending on the water vapour amount and the saturation of channels; hence, there are three "sub-algorithms" or retrieval regimes. Each sub-algorithm reaches its upper retrieval limit when the channel which is most sensitive to water-vapour becomes saturated. In the original algorithm formulation by Melsheimer and Heygster (2008), the switch from one sub-algorithm to the next (always starting

with the most sensitive one) is done only when the saturation condition,

$$T_{bj} - T_{bk} > 0 \tag{12}$$

is fulfilled. This means that for each satellite footprint, only one of the three sub-algorithms is finally used. As the sub-algorithms have been calibrated independently, the switch from one to the next can cause a jump in the retrieved value. A method avoiding this discontinuity in the retrieval values will be discussed further in the follow-on paper. Additionally, as the

switch between regimes is done in the brightness temperature space, this does not correspond to a strict cut-off point in water vapour. In Table 3 we summarize the characteristics of each regime.

## 2.6 Bias and calibration parameters

Since we ordered the channels by the water vapour sensitivity ($\tau_i < \tau_j < \tau_k$), the difference of exponentials in $\Delta T_{0ij}$ and $\Delta T_{0jk}$ is negative. Therefore, the first term of the temperature difference increases with increased emissivity from negative values to 0 (reached when $\epsilon = 1$). $\eta_c$ doesn't depend on $\epsilon$, which cancels on the ratioing. In a plot with $\Delta T_{jk}$ as abscissa and $\Delta T_{ij}$ as ordinate, for constant $W$ and varying $\epsilon$, this is a straight line with slope $\eta_c$ ($W$), running through the bias points $(b_{jk}, b_{ij})$. Since the biases depend only weakly on $W$ and $\epsilon$, all straight lines for different $W$ run through almost the same point $F = (F_{jk}, F_{ij})$, which is called focal point by Miao et al. (2001) and Melsheimer and Heygster (2008). The focal point $F$ is found by simulating brightness temperatures for a set of different $\epsilon$, with different input atmospheric profiles (including $W$) from radiosonde data, and surface temperature taken as ground-level atmospheric temperature (which makes the small emissivity dependence of the biases vanish; see Melsheimer and Heygster (2008) - Appendix II). Having determined the focal point, the simulated brightness temperature differences and corresponding TWV values from the radiosonde profiles can be used to get the calibration parameters $C_0$ and $C_1$. Thus, together with the two focal point coordinates $F_{jk}$ and $F_{ij}$, there is a total of four calibration parameters in the retrieval equation which are derived by this regression. The specific values for each viewing angle and regime of AMSU-B sensor are found in Melsheimer and Heygster (2008) – Appendix III. For MHS, all these calibration parameters were recalculated and are shown in Appendix A.

## 2.7 Filtering ice cloud artefacts

The effect of ice clouds at the AMSU-B frequencies as studied in Sreerekha (2005) is known, and has been used for detecting tropical deep convection (Hong et al., 2005) and for an automated method for finding polar mesocyclones (Melsheimer et al., 2016). The latter method uses the sensitivity of retrieved TWV to convective clouds with high ice content as one of the main signatures of polar lows. In these cases, since cloud ice particles are strong scatterers in the used microwave range, the radiation from below the clouds is scattered strongly and hardly reaches the sensor, so that the AMSU-B retrieval is only sensitive to atmospheric water vapour above such clouds and retrieves erroneously low TWV. A procedure to recognize and screen such cases for the AMSU-B/MHS algorithm has been developed. Cloud ice contents high enough to affect our TWV retrieval are almost entirely caused by strong convective clouds which are typically organised in rather small-scale (tens of kilometers) cells or clusters thereof, or which take the shape of mesoscale structures such a polar lows with extents of at most a few hundred kilometers; even in large scale, synoptic low pressure systems, convective clouds are organised in clusters and lines with the above-mentioned scales of tens to a few hundred kilometers. Therefore, image processing methods that rely on the size of ice cloud artefacts can be used: Our approach for eliminating the affected TWV is to find connected areas – minimum of two pixels – of low TWV ($<4 \, \mathrm{kg/m^2}$) smaller than 50 pixels which are surrounded by higher or non-retrieved values. The threshold of 50 pixels was selected because – with the data on the selected latitude-longitude grid of $0.25°$ – it would approximate to areas of $7000 \, \mathrm{km^2}$ at $60°$ N and $19600 \, \mathrm{km^2}$ at $80°$ N, and it amply covers the scale of events that need masking. Then, we remove these connected areas with a succession of morphology operations (Gonzalez and Woods, 2007), using the tools for Python described in van der Walt et al. (2014): First a dilation with a 7x7 square structural element, and then a closing with the

same size structural element. We ensure that only the data within the original connected areas are removed by using an image comparison between the mask and the initial connected areas.

## 3 Evaluation of retrieval

In this section, the performance of the TWV retrieval using MHS data is evaluated in subsection 3.1. Then, the satellite-based retrieval is firstly compared with in-situ data in subsection 3.2, and secondly to ERA5 reanalysis data in subsection 3.3.

### 3.1 Comparison between MHS and AMSU-B based retrieval

As shown in Table 1, there are some frequency and polarization differences between AMSU-B and MHS sensors. According to the analysis in John et al. (2012), there are some non-negligible discrepancies between the brightness temperatures of AMSU-B and MHS for the second and fifth channels (17 – 150 GHz – and 20 – $183.31 \pm 7$ GHz – for AMSU-B, respectively), due to the differences in frequency, while the differences in polarization seem not to be relevant. That raises the question of whether the TWV algorithm will perform equally when using MHS data as input, and, if that is not the case, which adaptation would be needed to ensure consistency of the retrieval results. One main adjustment we did to the retrieval for MHS is the recalculation of all the calibration parameters as described in Section 2.6 and shown in Appendix A.

First, we evaluate the performance for the retrieval as a whole by comparing the retrieved data of both algorithms in the overlap period of both sensors (2008-2009). For this analysis, we considered all the coincident points in the daily gridded data with a $0.25°$ grid. Figure 1 shows two density plots for the overlap months of January (top) and July (bottom) of 2008-2009. The results of a least squares regression are shown in the Figure as well. Both data sets show good agreement, with most of the points along the one-to-one line. However, we can observe some outliers with high MHS TWV and low, almost constant, AMSU-B TWV, and vice versa, specially striking during the month of July. These points are mostly associated with time differences of the satellite overpasses, and amount to only about 0.27% of the data, so they are not significant in the overall picture.

In Table 4, the fit statistics for all months are shown. The correlation ranges from 0.87 in June to 0.94 in September. The lowest slope (0.82) is found in December. On the other hand, the slope is closest to 1.0 in May (0.91). The intercept increases for the summer months (June, July, August) but is relatively small for the other months. The RMSD has a similar behaviour: we find higher values for the central months of the year, with a maximum of 2.25 kg/m$^2$ in August, coinciding with the increased number of outliers. Minimum is of 0.73 kg/m$^2$ in March. The bias is generally small (minimum of 0.04 kg/m$^2$ in March, maximum of 0.49 kg/m$^2$ in September), and positive except for May and June. In general, all parameters show lowest agreement in the summer months when the atmospheric variability is highest. However, we presume the strongest contribution to the lower agreement in summer is due to the higher uncertainty and variability in the surface emission due to melt process and occurence of melt ponds.

To check any possible influence from the surface type in the consistency of our retrievals, we have studied the TWV time series during 2008-2009 for MHS and AMSU-B over different surfaces: ice, land and open water. The location chosen for

each study point is shown in Figure 2, with the surface classification used in the TWV retrieval for a day in early March 2008 (maximum ice extent) as background. We show the monthly and yearly means of this time series for the four different locations in Figure 3. Note the lack of data for summer months over open water and ocean because of the limitations of the algorithm. All four time series show good agreement which confirms the consistency between our retrievals. The bias and RMSD are small for all four surface types (ice: $0.1\pm0.4$ kg/m$^2$, open water:$0.03\pm0.15$ kg/m$^2$, marginal ice zone: $0.2\pm0.7$ kg/m$^2$, land: $0.12\pm0.19$ kg/m$^2$), but slightly higher in two cases with ice surfaces, which agrees with the higher error of our method for higher water vapour values (extended regime).

## 3.2 Comparison with in-situ data sources

While TWV retrieved from AMSU-B has been validated with different data sources (Rinke et al., 2009; Palm et al., 2010; Buehler et al., 2012), the same cannot be said about the retrieval with MHS data. Therefore, we perform a comparison with TWV derived from radiosondes taken during the N-ICE21015 campaign from January to June 2015 onboard research vessel Lance north of Svalbard (Hudson et al., 2017; Cohen et al., 2017). We select the MHS data as the mean of all the values in a 50 km radius around the location of each radiosonde. The resulting time series is shown in Figure 4. The first thing to note is that the MHS series ends at the start of June because, afterward, the surface in the area is considered mixed according to the criteria described in Section 2.1. However, both data sets show good visual agreement, except that MHS is not able to capture some of the quasi periodic peaks in TWV from N-ICE2015 data set (seen roughly every two weeks in February and March). We have eliminated these nine outliers associated with the quasi periodic peaks in TWV from the following analysis. The scatter plot of all overlapping points of both data sets – with the colour scale representing the month of the campaign – shown in Figure 5, confirms the good agreement.

Additionally, we used Global Positioning System (GPS) and radiosounding (RS) TWV observations during the common 2008-2009 period between the AMSU-B and MHS sensors to evaluate the satellite TWV retrieval. GPS and radiosonde TWV have been measured at the five coastal Arctic stations Alert, Eureka, Ny Ålesund, Resolute and Scorebysund, as shown in Figure 6. These datasets are part of a homogenized time series. From the GPS data, 1-h average values of local integrated TWV have been computed each 6 hours. The radiosoundings have been performed once or twice per day at the selected sites (00:00 and 12:00 UTC). Further details about processing can be found in Negusini et al. (2016). As for the AMSU-B and MHS TWV values, we selected points fulfilling the data conditions of $\pm1$h from the integrated GPS measurements (00:00, 06:00, 12:00, 18:00UTC) and found in a 50 km radius around the GPS/RS stations. Additionally, TWV data from ERA5 reanalysis (Copernicus Climate Change Service , C3S) were obtained using the same conditions. The resulting AMSU-B, MHS, ERA5, GPS and radiosonde time series in Figure 7 present generally consistent patterns and reasonable seasonal evolution, with drier winters and wetter summers. Overall, the datasets have worse agreement during the summer months, mainly due to "spikier" data, i.e. more extreme water vapour values. Due to this pronounced seasonal cycle, we separate the results between summer (April to September) and winter (October to March) in the following analysis. There seems to be a slight wet bias in summer for both satellite-derived TWV with respect to the other datasets.

Scatter plots comparing each dataset (both satellite and reanalysis) with both radiosondes and GPS have been prepared for each season and station. As an example, Figure 8 shows the results for Alert. The correlation coefficients vary between 0.55 to 0.88, and the correlations in winter seems to be generally lower. We presume this is just a numerical effect because of the narrower data distribution. The RMSD, in contrast, is higher in summer (as seen in Figure 9). The only difference between both satellite-based retrievals seems to be a smaller number of coincident points between the MHS TWV and the radiosondes TWV (approximately half of the data points).

Figure 9 shows all fit parameters for the five stations, with separated results between summer and winter. There seems to be only little difference between the results from the two satellite-based retrievals, which corroborates our confidence in the MHS-based retrieval. Over the three quality indicating parameters RMSD, bias and correlation coefficient there is even a slight, but consistent advantage for the MHS based retrieval. The bias values are almost all negative, and the RMSD is along usual values for TWV studies at high latitudes (as seen in Palm et al. (2010) for Ny Ålesund and in Buehler et al. (2012) for Kiruna), which reassures us on the quality of satellite-based TWV retrievals. The higher RMSD values in the Arctic summer in Figure 9 can also be seen at high PW values over 7 kg/m$^2$ during summer for all methods in Figure 8 (left, top and bottom). One explanation for the smaller bias and RMSD during winter can be that also the absolute values during winter are small. The reason for a low correlation is likely that the temporal coherence is less pronounced.

When fits like in Figure 8 are performed for all stations for ERA5 versus GPS and radiosondes, the slopes are closer to one in summer (0.99 for GPS and 0.87 for radiosondes in average for all stations) but underestimate data to a higher degree in winter (in average, 0.85 for GPS and 0.76 for radiosondes). This seasonal variation is similar for the correlation coefficient, higher in summer – averaging 0.9 and 0.92 — and lower in winter, 0.85 and 0.75 for GPS and radiosondes, respectively. These values are very similar to the averages for the satellite data versus the in-situ data. The RMSD and bias are generally small, but smaller in winter. The average RMSD is 1.89 kg/m$^2$ in summer, 1.10 kg/m$^2$ in winter for GPS, and 1.58 kg/m$^2$ in summer and 1.05 kg/m$^2$ in winter for radiosondes. The average bias is generally negative for GPS, averaging -0.5 kg/m$^2$ in summer and -0.02 kg/m$^2$ in winter, while it is always positive for radiosondes, averaging 0.34 kg/m$^2$ in summer and 0.17 kg/m$^2$ in winter.

## 3.3 Comparison with ERA5 reanalysis

The reanalysis product ERA5 combines a variety of observations and a numerical model using an optimization procedure. Due to ERA5 assimilation of some of the observations used as verification here, namely radiosondes, it is not a completely independent estimate of TWV. ERA5 also assimilates some 183GHz data over sea ice and snow-covered surfaces (as suggested in Bormann et al., 2017), including the MHS sounding channels. While it is unclear to the authors which sensors would have been available and assimilated within ERA5 for the time period 2008-2010 of this study, we cannot presume that ERA5 TWV is entirely independent of microwave humidity sounder radiances.

For this study, we have compiled all the overlapping daily means of TWV from AMSU-B and ERA5 (Copernicus Climate Change Service , C3S) for the complete months of January (top) and July (bottom) from 2008 to 2009, shown in Figure 10. The results of a least squares regression are shown in the Figure as well. Both data sets show good agreement, with most of the points along or parallel to the one-to-one line. Low AMSU-B TWV values compared to high ERA5 TWV values can be

observed in both months, but are more prominent in summer. These are remnants of ice cloud artefacts that were not entirely filtered out.

Table 5 shows the fit statistics for all months. The correlation ranges from 0.71 in June to 0.88 in December. The worst slope (1.6) is found in September. On the other hand, the slope is closest to 1.0 in August (0.97). However, the RMSD has higher values for summer months of the year, with a maximum of 5.9 kg/m$^2$ in August, coinciding with the increased number of outliers. Minimum is of 1.00 kg/m$^2$ in March. The bias is generally negative, and shows similar behaviour as the RMSD. In general, all parameters show lowest agreement in the summer months when the atmospheric variability is highest.

## 4 Evaluation of changes/improvements in the retrieval: Filtering ice cloud artefacts

Figure 11 shows daily averaged TWV maps – with the ice cloud filtering (Section 2.7) already applied – for the AMSU-B/MHS algorithm (top and second row), as well as from a different data product based on AMSR-E observations (Wentz and Meissner, 2006) over open ocean (third row) and ERA5 reanalysis daily mean (bottom) in winter (left) and summer (right). The days chosen to represent each season (6 January and 6 July, 2008, respectively) show how a typical retrieval looks like for the respective season. The first thing to notice is the difference in spatial coverage of AMSU-B TWV between winter and summer. In summer, AMSU-B/MHS retrieval is restricted to the drier regions, mostly over sea ice and Greenland (the upper limit of the retrieval is usually about 15 kg/m$^2$ for sea ice surfaces). In winter, the retrieval is possible over most of the land, open water areas and sea ice. Meanwhile, there is no significant coverage variation shown between seasons for the AMSR-E retrieval: most open water areas are covered. In consequence, the area covered by both methods is smaller in summer, as we can note in the map illustrating the regional coverage – for the same days – of both algorithms in Figure 12 (orange area shows joint coverage). Still, TWV is retrieved in most of the Arctic in both seasons. Another consequence is that in summer the overlap area is small. In this particular example of Figure 12, there is no overlap between both datasets. As for the ERA5 dataset, the agreement with both AMSU-B and AMSR-E is qualitatively good, showing similar patterns, particularly in winter.

To visualize the areas affected by the ice cloud artifact, Figure 13 shows different areas of interest before (left) and after (right) filtering, for different days spaced evenly throughout 2008 (each three months approximately: 6th of January, 2nd of April, 6th of July and 14th of October). These areas have been chosen as representative cases for the season. Most features – small regions of low TWV surrounded by high TWV – are removed, but there are still some small areas of low values of TWV (such as the retrieved regions in the land around 70° W, 62°N Figure 13 (October, bottom right)). Note that these incorrectly retrieved areas are surrounded by grey values which represent water vapour too high to be retrieved with the AMSU-B method (about >7 kg/m$^2$ over ocean or land surfaces). We confirmed by comparison to the ERA5 atmospheric reanalysis that the remaining high TWV values are within the expected range. Also the high, >14 kg/m$^2$, TWV values on 6th July 2008 in the Hudson Bay area are in agreement with ERA5.

To show the overall effectiveness of the ice cloud filtering, we have compiled all the overlapping retrieved TWV from AMSU-B and AMSR-E for the complete months of January (top) and July (bottom) from 2006 to 2008, shown in Figure 14. Before filtering for ice cloud artefacts (left), there is a big cluster of data with high AMSR-E values for relatively low AMSU-B

values. Those correspond to the values affected by convective clouds with high ice content. Note that the overlap area between AMSR-E and AMSU-B is small (Figure 12) and therefore cloud artefacts make up a large fraction of the overlap data points, particularly in summer. After filtering (Figure 14, right) the AMSU-B retrieval, they are gone. Additionally, the fit performed improves significantly, with the correlation reaching 0.6 in summer and the slope getting much closer to one in winter (0.95, as compared to 0.3). Note also the jump in density of the retrieved TWV values caused by switching between sub-algorithms mentioned above (Section 2.5), most notably near 6 kg/m$^2$ (Figure 14). Between 7.6% (January) and 11% (March) of the data are masked by the ice cloud filter for winter months, while the percentage is much smaller in the summer months, ranging between 0.18% of the data in August to 3.7% in June. In summer, up to 94% of those values (July) come from the overlap area between AMSU-B and AMSRE, with the average 55.5%. In winter, the values from the overlap area average 11.8%.

## 5   Conclusions

We provide an updated version of the TWV retrieval algorithm that originally uses as input microwave humidity sounder data from AMSU-B. The updated algorithm, can now also use data from MHS, the successor instrument of AMSU-B, and contains a filter for artefacts caused by convective clouds with high cloud ice content. The improved retrieval performs better when compared to another satellite product and to in situ data.

The coefficients in the retrieval algorithm were adapted for MHS (Appendix A). We have investigated the impact of differences between AMSU-B and MHS on the retrieved TWV and have found the differences to be negligible. This means that a consistent continuous data set for the years 1999 until now can be generated from combining AMSU-B and MHS data. Additionally, the MHS-based TWV data have been compared with radiosonde data from the N-ICE2015 campaign, and the results show good performance for MHS TWV. Both satellite-derived TWV have been compared against GPS and radiosonde data for five Arctic coastal stations during 2008 and 2009, and the results are satisfactory, with averaged correlations for all stations and methods 0.82 in summer and and 0.75 in winter, and RMSD along usual values for TWV studies at high latitudes. The satellite based TWV retrieval also compares well with the ERA5 reanalysis. Some artefacts of not filtered ice clouds remain but overall the correlation with 0.79 and RMSD of 3.01 kg/m$^2$ show good correspondence.

The filter for ice cloud artefacts performs well as shown by comparison with data from the AMSR-E based algorithm that works over open water. A remaining issue are the jumps of retrieved TWV values between the different retrieval regimes. This can, however, in principle be mitigated by comparing root mean square differences and bias for adjacent TWV regimes, and choosing an optimal regime, i.e., channel combination, for the range of the water vapour column. Where regimes overlap, weighted averages can smooth the transition.

The algorithm described here has an upper TWV limit that restricts retrieval in summer to the central Arctic and Greenland. However, when combining the TWV data retrieved by the algorithm described here with TWV retrieved over open ocean from AMSR-E and AMSR2 – the product by Remote Sensing Systems (RSS) (Wentz and Meissner, 2006) – a nearly complete coverage of the whole Arctic year-round is possible, starting in 2000, which is the overall goal of future work.

## Appendix A

The following Tables list the calibration parameters $C_0$, $C_1$, $F_{jk}$, and $F_{ij}$ for the TWV retrieval algorithm for the Arctic and – for the sake of completeness – the Antarctic, for 15 viewing angles that span the range of the viewing angles of MHS, calculated in the same way as the parameters for AMSU-B-based retrieval by Melsheimer and Heygster (2008). The retrieval equation is, from (5) and (8):

$$W \sec\theta = C_0 + C_1 \ln\left[\frac{\Delta T_{ij} - F_{ij}}{\Delta T_{jk} - F_{jk}}\right], \tag{A1}$$

where $\Delta T_{ij} = T_{b,i} - T_{b,j}$, the MHS channels $i, j, k$ are

- 5 (190.31 GHz), 4 (183.31 $\pm$ 3 GHz), 3 (183.31 $\pm$ 1 GHz) for the low-TWV algorithm,

- 2 (157 GHz), 5 (190.31 GHz), 4 (183.31 $\pm$ 3 GHz) for the mid-TWV algorithm,

and, from equations ( 10) and ( 9),

$$W \sec\theta = C_0 + C_1 \ln\left[\frac{r_j}{r_i}\left(\frac{\Delta T_{ij} - F_{ij}}{\Delta T_{jk} - F_{jk}} + 1.1\right) - 1.1\right] \tag{A2}$$

where $i,j,k$ are 1 (89.9 GHz), 2 (157 Ghz), 5 (190.31 GHz) for the extended algorithm.

The calibration parameters for the Arctic (Tables A1–A3) were derived using radiosonde data from those World Meteorological Organization (WMO) stations in the Arctic that are located on the coast or on islands(29 stations), from the years 1996 to 2002, which amounts to about 27000 radiosonde profiles.

*Competing interests.* No competing interests are present

*Acknowledgements.* AMSR TWV data are produced by Remote Sensing Systems and were sponsored by the NASA AMSR-E Science Team and the NASA Earth Science MEaSUREs Program. Data is available at www.remss.com.

We gratefully acknowledge the funding by the Deutsche Forschungsgemeinschaft (DFG, German Research Foundation) Project number 268020496 - TRR 172, within the Transregional Collaborative Research Center "ArctiC Amplification: Climate Relevant Atmospheric and SurfaCe Processes, and Feedback Mechanisms, (AC)[3], as well as the support by the project INTAROS (INTegrated Arctic Observation System) funded by the European Union's Horizon 2020 Research and Innovation Programme under GA 727890.

The authors would like to thank the Department of Atmospheric Science, University of Wyoming (http://weather.uwyo.edu/upperair/sounding.html) and the N-ICE2015 campaign (www.npolar.no/en/projects/n-ice2015/) for the radiosonde raw data, the International GNSS Service (ftp://igs.ensg.ign.fr/pub/igs/data/) for the GPS data and Copernicus Climate Change Service for the ERA5 data.

We thank the three reviewers for their helpful comments, which helped to significantly improve the manuscript.

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

**Figure 1.** Density plot and fit for MHS TWV versus AMSU-B TWV retrievals for all the coincident points in January (top) and July (bottom), 2008-2010. The dashed line is the one-to-one line, and the black line corresponds to the linear fit of the data.

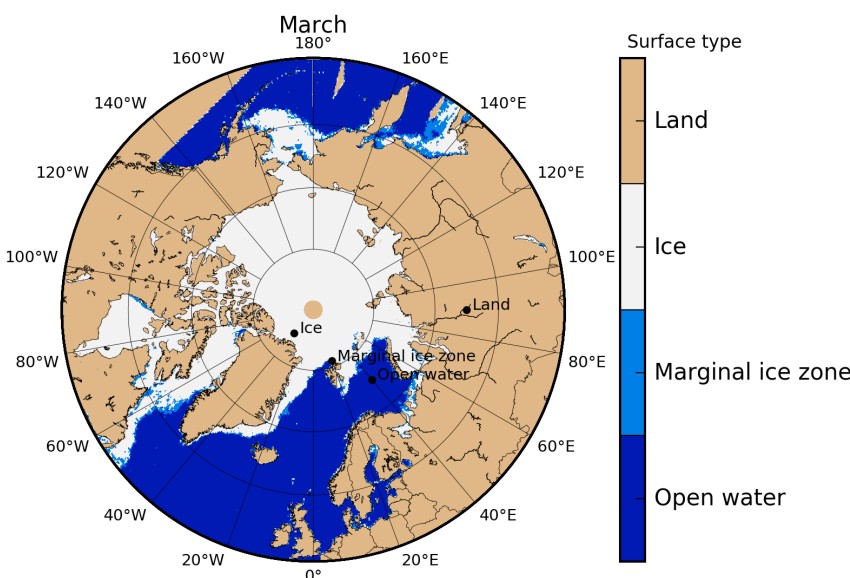

**Figure 2.** In black, location of the points chosen for the surface characterisation study for TWV. As background, the surface classification used in the TWV algorithm, obtained from ASI algorithm ice concentration (Spreen et al., 2008) for a typical day in March (6.03.2008).

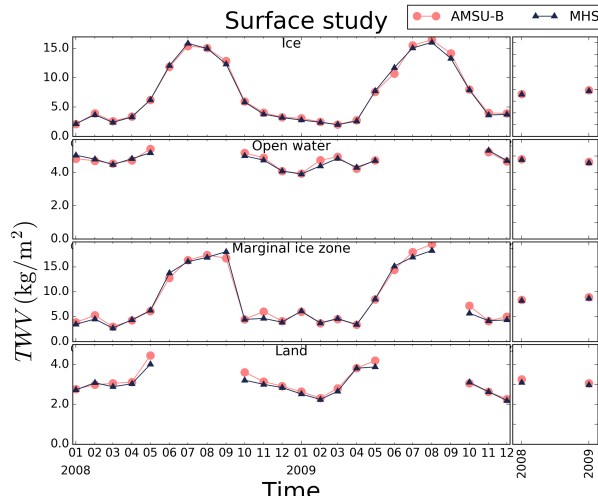

**Figure 3.** Monthly and yearly means for 2008 and 2009 of the AMSU-B (pink circles) and MHS (blue triangles). TWV retrieval over the locations shown in Figure 2.

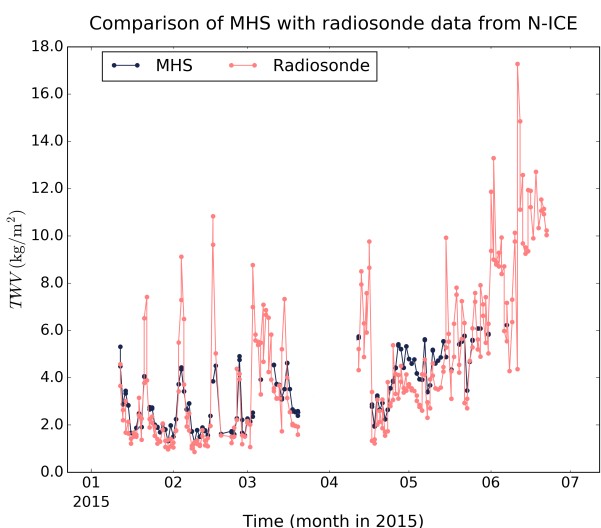

**Figure 4.** Time series of coincident MHS TWV data (blue symbols) and TWV from radiosondes (red symbols) during the N-ICE campaign.

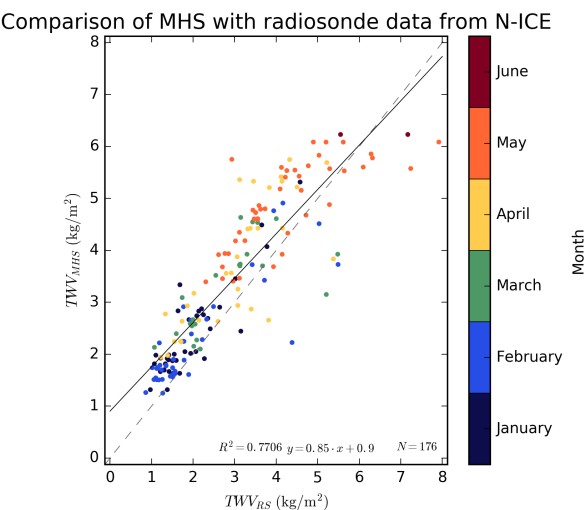

**Figure 5.** Scatter plot and fit for MHS TWV versus radiosonde TWV retrievals for all coincident points during the N-ICE campaign. The colour scale shows the month where the data point comes from; dashed line: one-to-one lines, solid line: linear regression.

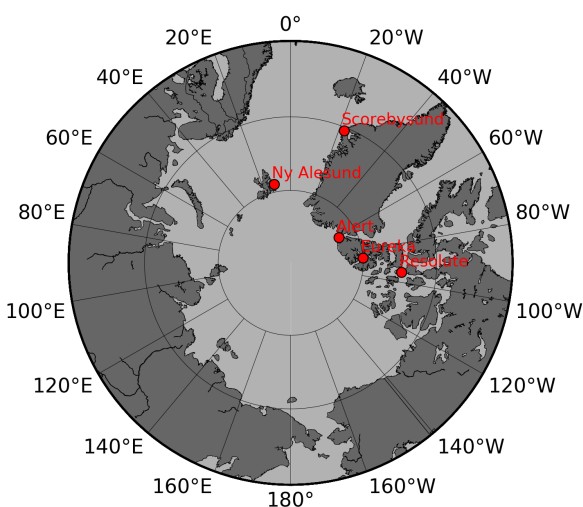

**Figure 6.** Location of the radiosonde and GPS stations.

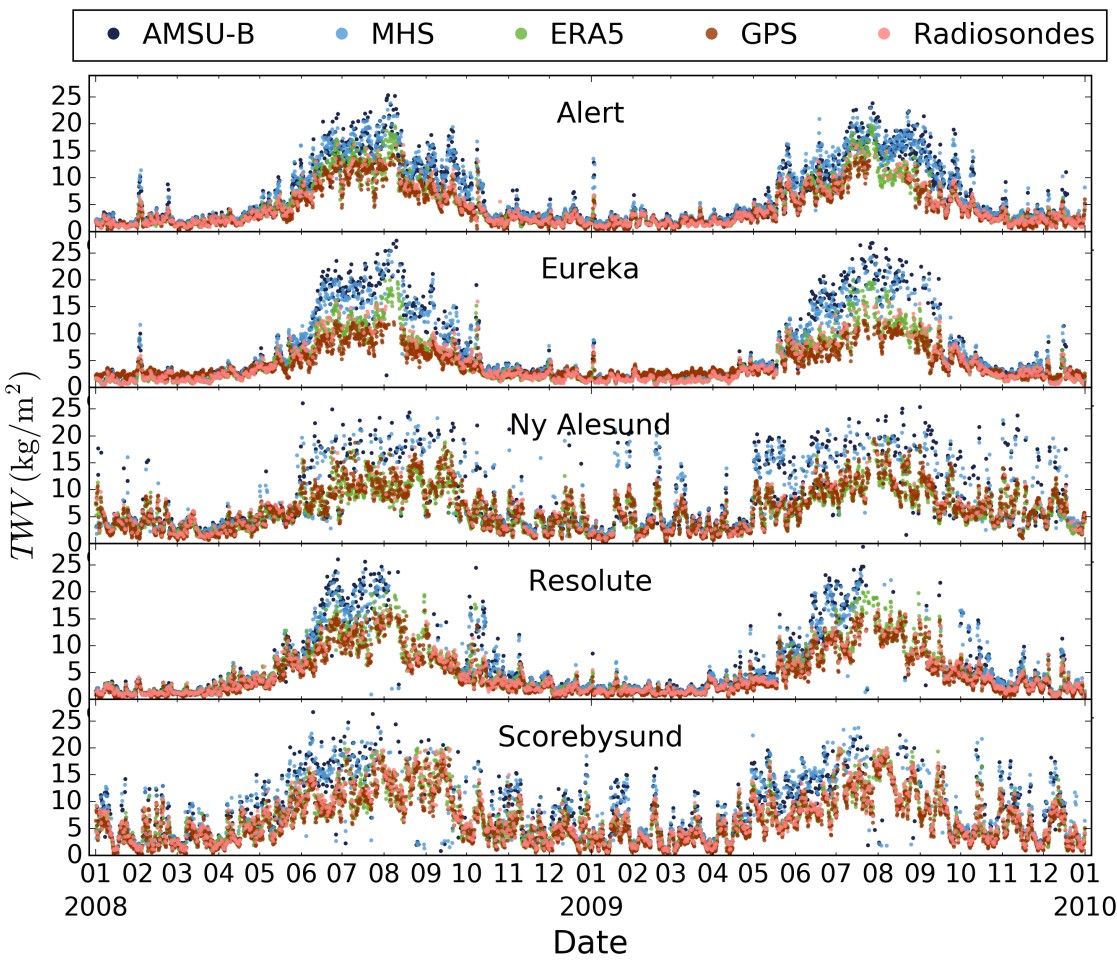

**Figure 7.** Time series of AMSU-B (dark blue), MHS (light blue), ERA5 (green), GPS (purple) and radiosonde (salmon) TWV retrievals during 2008 and 2009.

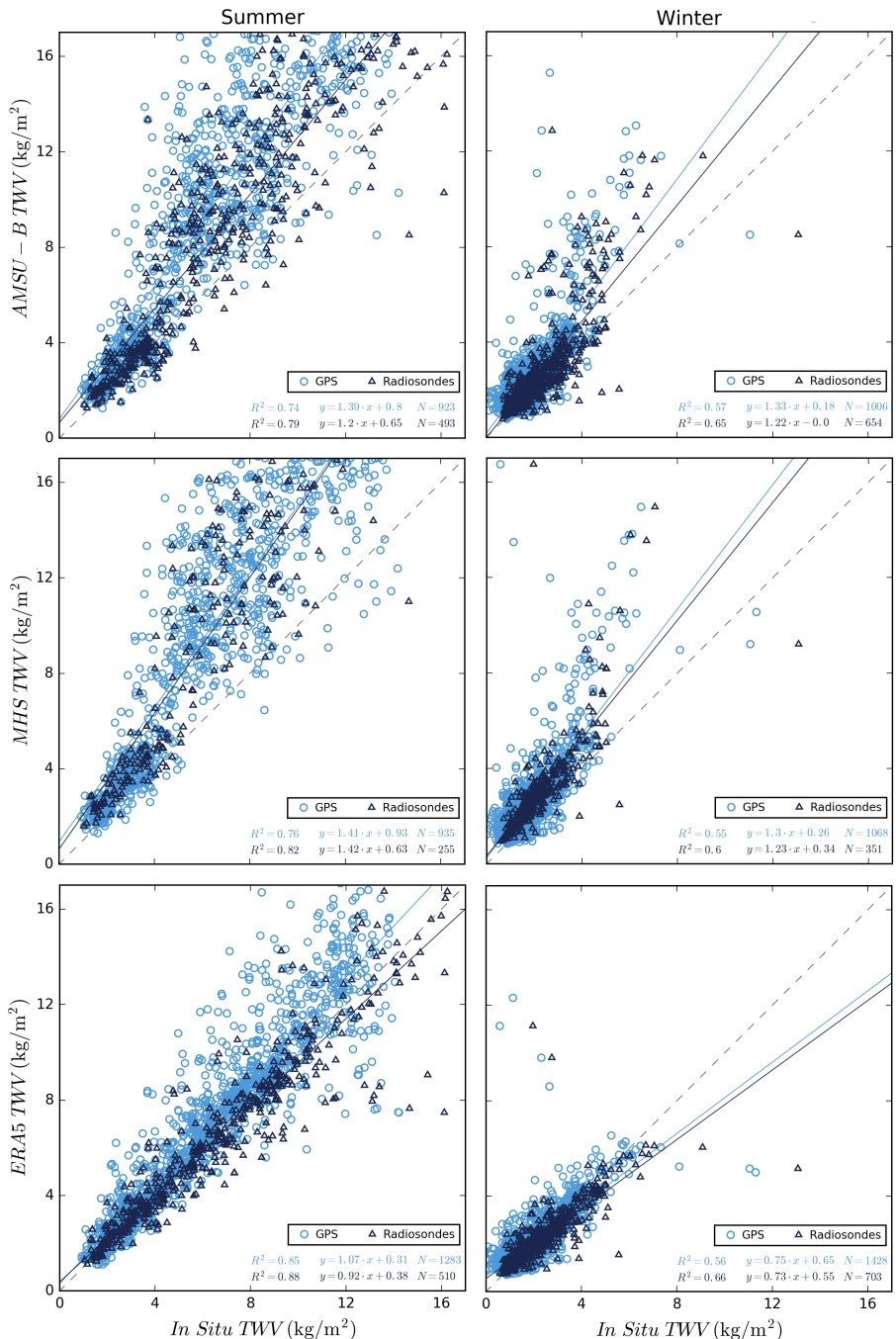

**Figure 8.** Scatter plots and fits for AMSU-B (top) and MHS (middle) and ERA5 (bottom) TWV retrievals versus GPS (light blue) and radiosondes (dark blue) TWV retrievals for all coincident points during summer (left) and winter (right) of 2008 and 2009 in the Alert station. The solid lines in light and dark blue show the linear regressions for GPS and radiosondes in each case, while the dashed lines are the identity line

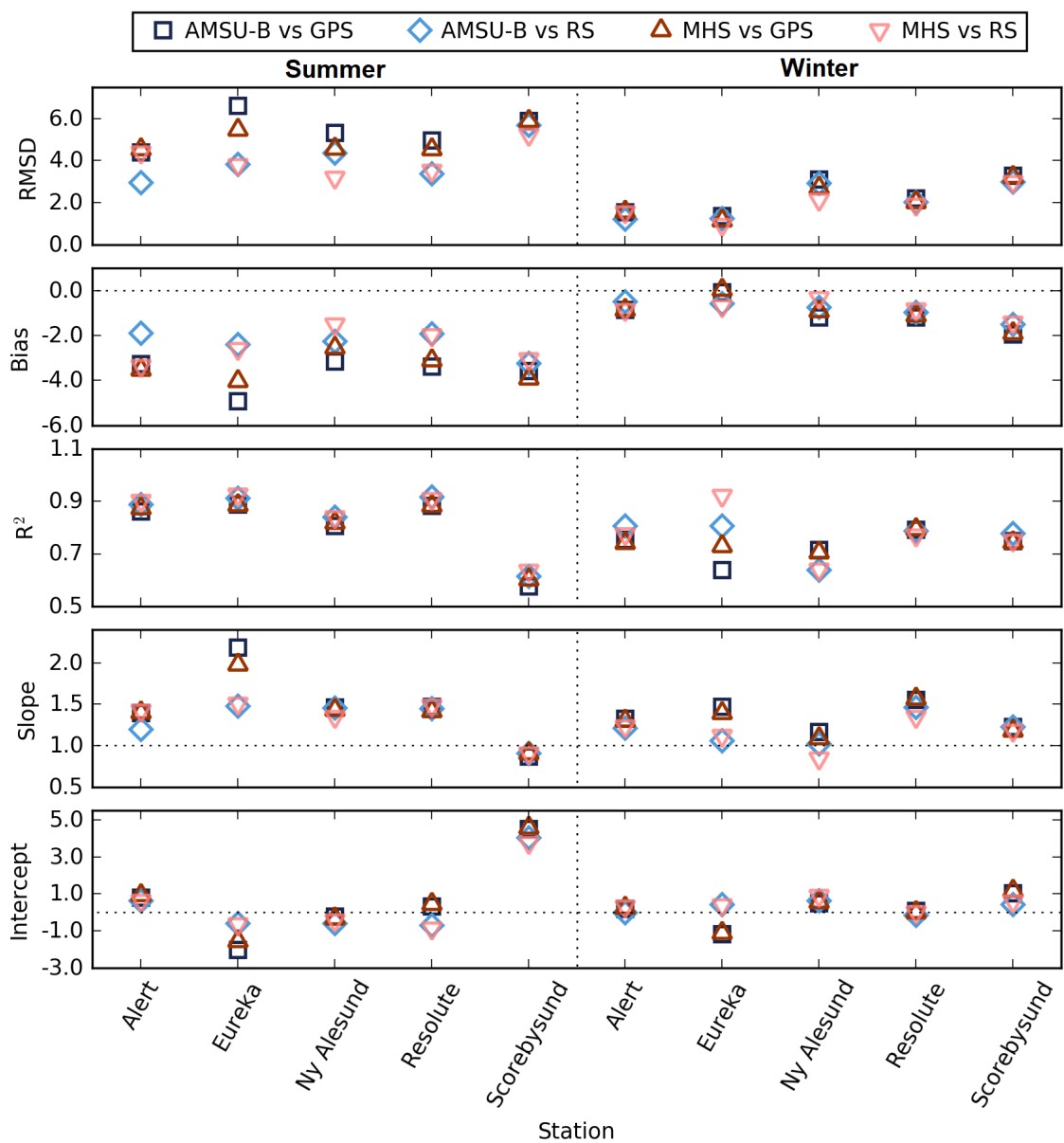

**Figure 9.** Values of fit parameters for summer (left) and winter (right): RMSD, bias, correlation coefficient $R^2$, slope and intercept of regression line for MHS and AMSU-B TWV retrievals versus radiosonde and GPS TWV retrievals. RMSD, bias and intercept are in kg/m$^2$; slope and $R^2$ are absolute numbers.

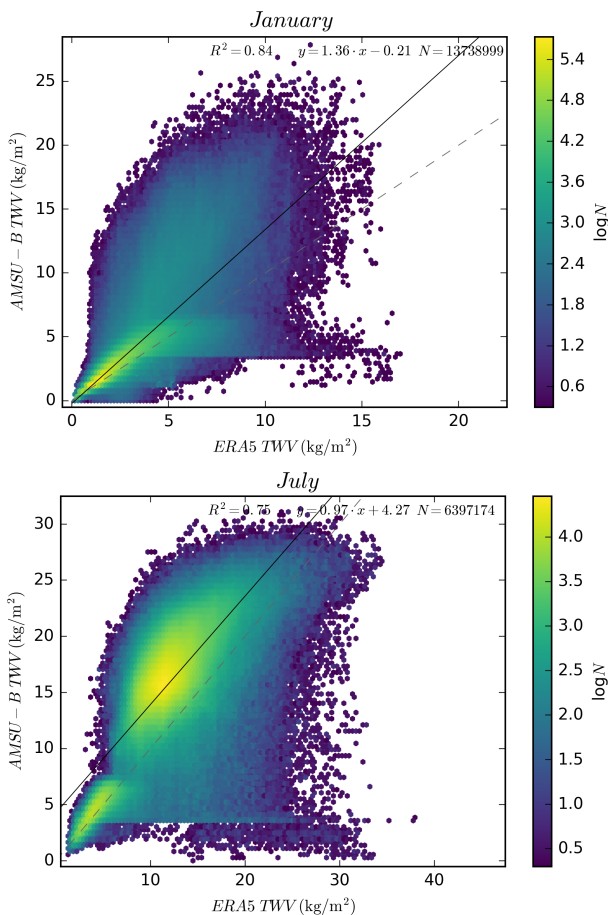

**Figure 10.** Density plot and fit for AMSU-B TWV versus ERA5 TWV retrievals for all the coincident points in January (top) and July (bottom) from 2008 to 2009, with a fit (black solid line) for the data clusters over the 1-1 line (dashed grey).

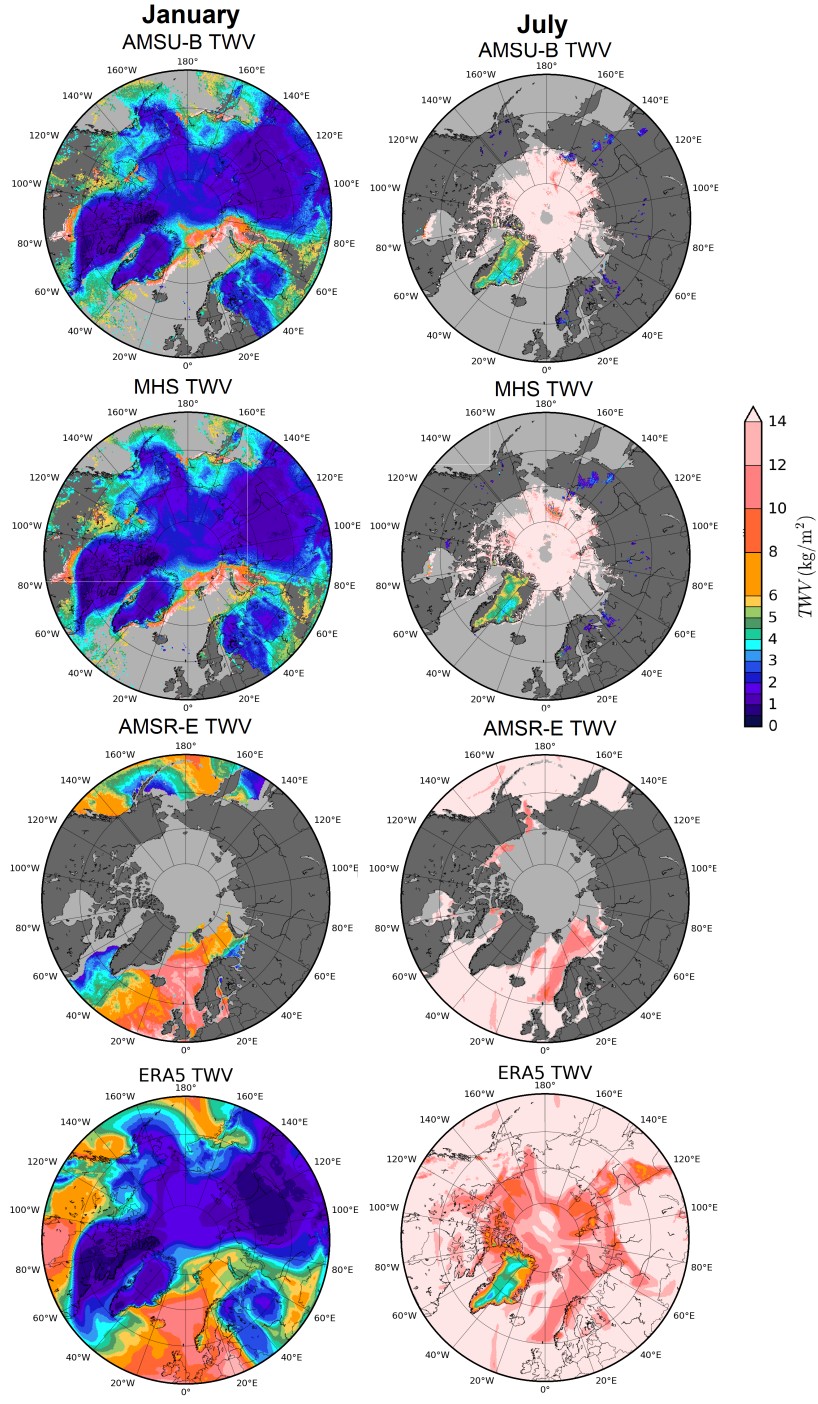

**Figure 11.** AMSU-B (top), MHS (second row), AMSR-E (third row) and ERA5 (bottom) TWV retrievals for (left) winter (6 January 2008), and (right) summer (6 July 2008)

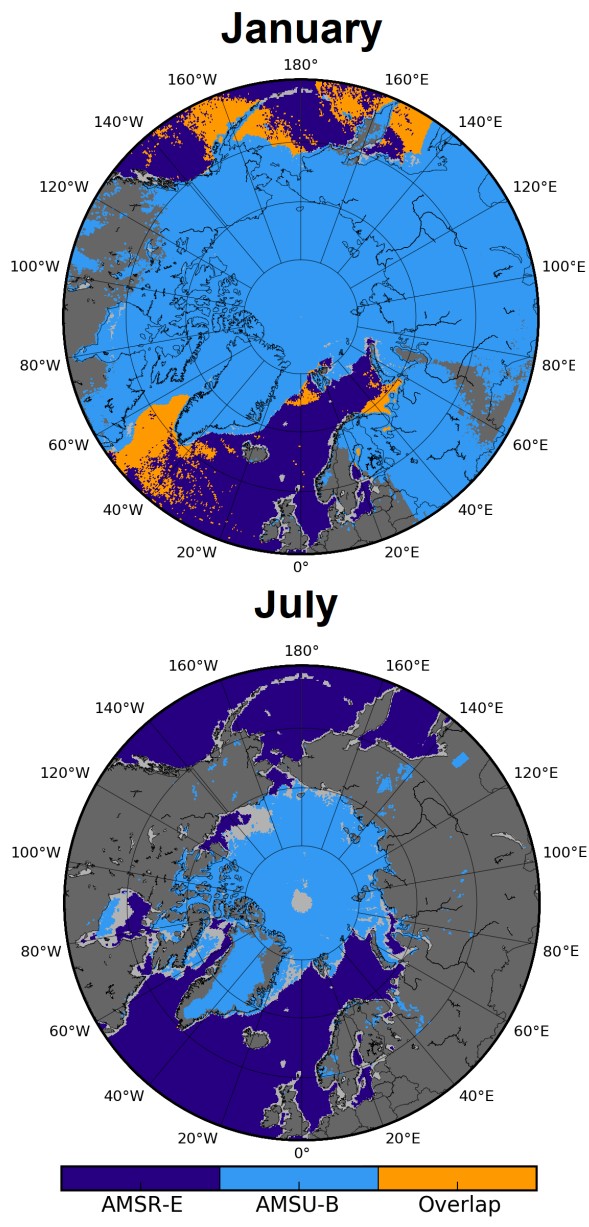

**Figure 12.** Coverage and overlap area of the merged AMSU-B and AMSR-E retrieval for (top) winter (6 January, 2008), and (bottom) summer (6 July, 2008). Note that there is no overlap between retrievals (orange) for the summer case presented.

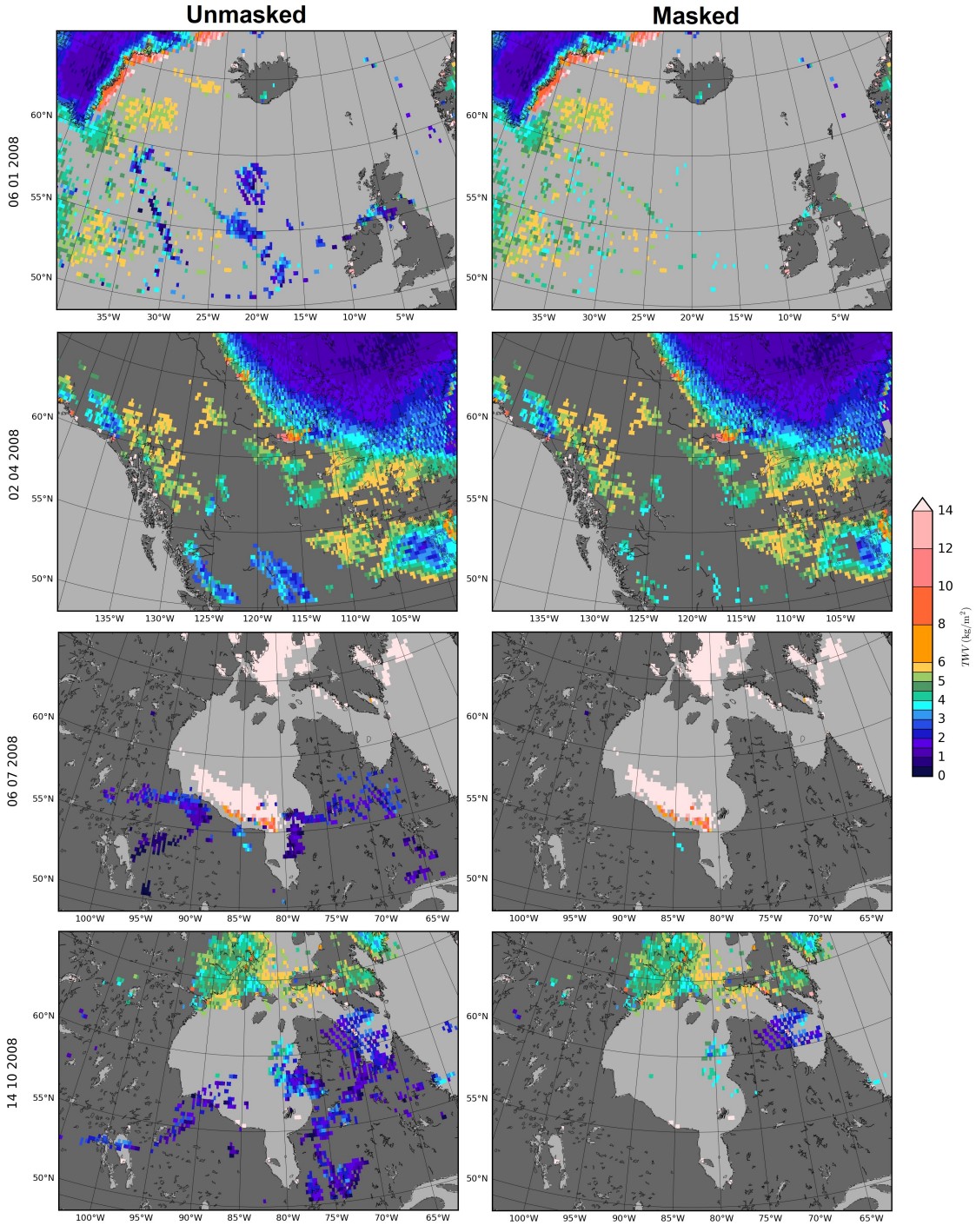

**Figure 13.** Unmasked (left) and masked (right) AMSU-B TWV retrieval for different showcased areas of four days through 2008: 6 January (top), 2 April (middle up), 6 July (middle down) and 14 October (bottom). Please note the different location in each case.

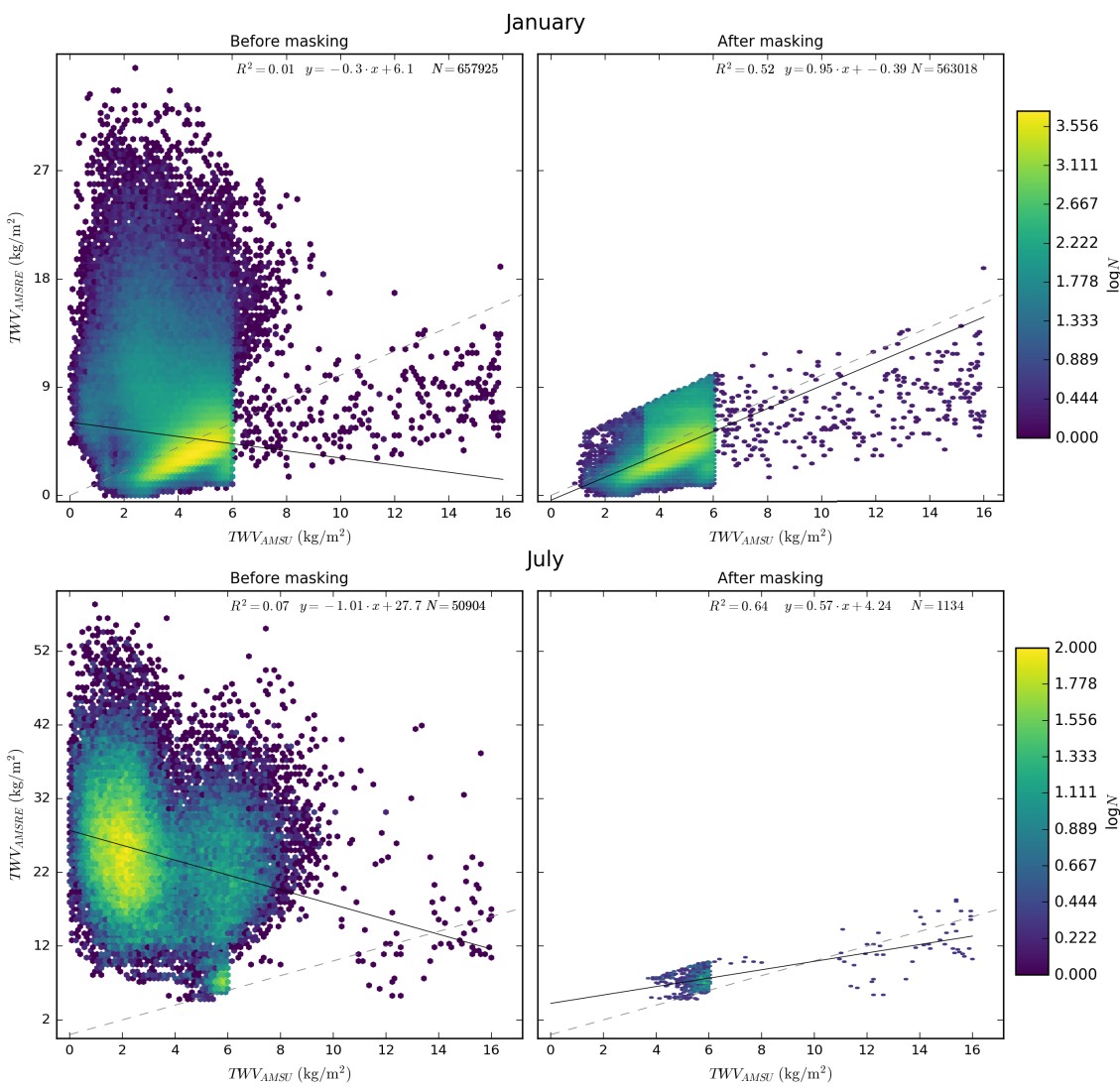

**Figure 14.** Density plot and fit for AMSR-E TWV versus AMSU-B TWV retrievals for all the coincident points in January (top) and July (bottom) from 2006 to 2008, before (left) and after (right) filtering AMSU-B retrieval for ice cloud artefacts, with a fit (black solid line) for the data clusters over the 1-1 line (dashed grey).

**Table 1.** Frequency and polarization details for each channel of AMSU-B and MHS sensor

| AMSU-B | | | MHS | | |
|---|---|---|---|---|---|
| Channel | Frequency (GHz) | Polarisation | Channel | Frequency (GHz) | Polarisation |
| 16 | $89.9 \pm 0.9$ | Vertical | 1 | 89.9 | Vertical |
| 17 | $150.0 \pm 0.9$ | Vertical | 2 | 157.0 | Vertical |
| 18 | $183.31 \pm 1$ | Vertical | 3 | $183.31 \pm 1$ | Horizontal |
| 19 | $183.31 \pm 3$ | Vertical | 4 | $183.31 \pm 3$ | Horizontal |
| 20 | $183.31 \pm 7$ | Vertical | 5 | 190.311 | Vertical |

**Table 2.** Humidity Sounders in orbit with platforms, launch year and approximate equator crossing times (ECT) [NOAA]

| Platform | Sensor | Launch year | ECT |
|----------|--------|-------------|------|
| NOAA15 | AMSU-B | 1999 | 07:00 |
| NOAA16 | AMSU-B | 2000 | 21:00 |
| NOAA17 | AMSU-B | 2002 | 07:00 |
| NOAA18 | MHS | 2005 | 20:00 |
| NOAA19 | MHS | 2009 | 20:00 |
| MetOp-A | MHS | 2006 | 9:30 |
| MetOp-B | MHS | 2012 | 9:30 |
| MetOp-C | MHS | 2018 | 9:30 |

**Table 3.** Characteristics of the different sub-algorithms of the AMSU-B/MHS TWV retrieval. The channel combination is described with AMSU-B frequencies, the MHS retrieval uses the corresponding ones

| Sub-algorithm | Chanel combination | | | Operating surface | Approximate limit TWV (kg/m$^2$) |
|---|---|---|---|---|---|
| Low | $183.31 \pm 7$ | $183.31 \pm 3$ | $183.31 \pm 1$ | Sea ice, ocean, land | 1.5 |
| Middle | $183.31 \pm 7$ | $183.31 \pm 3$ | 150 | Sea ice, ocean, land | 7 |
| Extended | $183.31 \pm 7$ | 150 | 89 | Sea ice | 15 |

**Table 4.** Parameters for linear regression for monthly MHS and AMSU-B intercomparison

| Month | $R^2$ | Slope | Intercept (kg/m$^2$) | RMSD (kg/m$^2$) | Bias (kg/m$^2$) | Number of points |
|---|---|---|---|---|---|---|
| January | 0.90 | 0.85 | 0.37 | 0.97 | 0.06 | 10691385 |
| February | 0.89 | 0.84 | 0.38 | 0.87 | 0.05 | 9858305 |
| March | 0.90 | 0.87 | 0.31 | 0.73 | 0.04 | 10389349 |
| April | 0.90 | 0.88 | 0.40 | 1.02 | 0.06 | 8592621 |
| May | 0.91 | 0.91 | 0.61 | 1.59 | -0.02 | 6087842 |
| June | 0.87 | 0.84 | 2.33 | 2.25 | -0.38 | 4741678 |
| July | 0.88 | 0.83 | 2.23 | 2.18 | 0.38 | 3803287 |
| August | 0.92 | 0.88 | 1.43 | 2.07 | 0.35 | 3272951 |
| September | 0.94 | 0.89 | 0.83 | 1.77 | 0.49 | 3630497 |
| October | 0.93 | 0.86 | 0.67 | 1.55 | 0.19 | 6000153 |
| November | 0.90 | 0.85 | 0.53 | 1.20 | 0.06 | 8610697 |
| December | 0.88 | 0.82 | 0.50 | 1.24 | 0.12 | 7723324 |

**Table 5.** Parameters for linear regression for monthly AMSU-B and ERA5 intercomparison

| Month | $R^2$ | Slope | Intercept (kg/m$^2$) | RMSD (kg/m$^2$) | Bias (kg/m$^2$) | Number of points |
|---|---|---|---|---|---|---|
| January | 0.84 | 1.36 | -0.22 | 1.43 | -0.58 | 13738999 |
| February | 0.85 | 1.33 | -0.2 | 1.25 | -0.52 | 12626117 |
| March | 0.85 | 1.17 | 0.06 | 1.00 | -0.44 | 14004549 |
| April | 0.84 | 1.16 | 0.12 | 1.38 | -0.61 | 11891174 |
| May | 0.78 | 1.25 | -0.19 | 2.72 | -1.21 | 8511599 |
| June | 0.71 | 1.11 | 1.69 | 4.32 | -2.72 | 7212136 |
| July | 0.75 | 0.97 | 4.28 | 4.89 | -3.87 | 6397174 |
| August | 0.73 | 0.97 | 5.1 | 5.91 | -4.79 | 5376590 |
| September | 0.81 | 1.54 | 0.95 | 5.83 | -4.77 | 5692249 |
| October | 0.74 | 1.67 | -0.24 | 3.54 | -2.11 | 9281562 |
| November | 0.77 | 1.5 | -0.41 | 2.1 | -1.05 | 11880942 |
| December | 0.88 | 0.82 | 0.50 | 1.24 | 0.12 | 10822902 |

**Table A1.** Calibration parameters, Arctic, low-TWV algorithm

| $\theta$ | $C_0$ [kg/m$^2$] | $C_1$ [kg/m$^2$] | $F_{4,3}^L$ [K] | $F_{5,4}^L$ [K] |
|---|---|---|---|---|
| 1.667° | 0.619 | 1.05 | 4.86 | 4.43 |
| 5.000° | 0.619 | 1.05 | 4.87 | 4.45 |
| 8.333° | 0.618 | 1.05 | 4.90 | 4.50 |
| 11.667° | 0.617 | 1.05 | 4.94 | 4.58 |
| 15.000° | 0.615 | 1.05 | 4.99 | 4.68 |
| 18.333° | 0.613 | 1.05 | 5.06 | 4.81 |
| 21.667° | 0.609 | 1.05 | 5.14 | 4.97 |
| 25.000° | 0.606 | 1.04 | 5.23 | 5.16 |
| 28.333° | 0.601 | 1.04 | 5.32 | 5.36 |
| 31.667° | 0.598 | 1.02 | 5.31 | 5.41 |
| 35.000° | 0.597 | 1.00 | 5.25 | 5.36 |
| 38.333° | 0.602 | 0.96 | 5.01 | 4.96 |
| 41.667° | 0.603 | 0.92 | 4.76 | 4.50 |
| 45.000° | 0.607 | 0.87 | 4.43 | 3.85 |
| 48.333° | 0.607 | 0.80 | 4.12 | 3.27 |

**Table A2.** Calibration parameters, Arctic, mid-TWV algorithm

| $\theta$ | $C_0$ [kg/m$^2$] | $C_1$ [kg/m$^2$] | $F_{5,4}^M$ [K] | $F_{2,5}^M$ [K] |
|---|---|---|---|---|
| 1.667° | 1.63 | 2.64 | 6.56 | 5.74 |
| 5.000° | 1.63 | 2.64 | 6.55 | 5.75 |
| 8.333° | 1.62 | 2.64 | 6.54 | 5.75 |
| 11.667° | 1.61 | 2.63 | 6.52 | 5.75 |
| 15.000° | 1.60 | 2.62 | 6.50 | 5.77 |
| 18.333° | 1.59 | 2.61 | 6.46 | 5.77 |
| 21.667° | 1.57 | 2.59 | 6.43 | 5.79 |
| 25.000° | 1.55 | 2.57 | 6.38 | 5.82 |
| 28.333° | 1.53 | 2.54 | 6.34 | 5.86 |
| 31.667° | 1.50 | 2.50 | 6.25 | 5.86 |
| 35.000° | 1.46 | 2.46 | 6.18 | 5.90 |
| 38.333° | 1.42 | 2.40 | 6.09 | 5.95 |
| 41.667° | 1.37 | 2.33 | 5.99 | 6.01 |
| 45.000° | 1.30 | 2.24 | 5.83 | 6.03 |
| 48.333° | 1.22 | 2.11 | 5.65 | 6.08 |

**Table A3.** Calibration parameters, Arctic, extended algorithm

| $\theta$ | $C_0$ [kg/m$^2$] | $C_1$ [kg/m$^2$] | $F_{2,5}^E$ [K] | $F_{1,2}^E$[K] |
|---|---|---|---|---|
| 1.667° | 14.4 | 7.45 | 6.52 | 0.74 |
| 5.000° | 14.4 | 7.47 | 6.55 | 0.74 |
| 8.333° | 14.4 | 7.50 | 6.61 | 0.75 |
| 11.667° | 14.4 | 7.56 | 6.71 | 0.77 |
| 15.000° | 14.4 | 7.63 | 6.84 | 0.80 |
| 18.333° | 14.4 | 7.73 | 7.00 | 0.83 |
| 21.667° | 14.5 | 7.83 | 7.20 | 0.87 |
| 25.000° | 14.5 | 7.97 | 7.44 | 0.93 |
| 28.333° | 14.5 | 8.11 | 7.72 | 1.00 |
| 31.667° | 14.5 | 8.26 | 8.04 | 1.08 |
| 35.000° | 14.5 | 8.43 | 8.41 | 1.19 |
| 38.333° | 14.4 | 8.60 | 8.83 | 1.33 |
| 41.667° | 14.2 | 8.76 | 9.30 | 1.50 |
| 45.000° | 13.9 | 8.90 | 9.83 | 1.74 |
| 48.333° | 13.4 | 8.99 | 10.4 | 2.04 |