# Peer review of "Improved Water Vapour retrieval from AMSU-B/MHS in the Arctic"

_Atmospheric Measurement Techniques, 2019_

## Referee Comment (RC1) · Anonymous Referee #1 · 9 Oct 2019

The retrieval of water vapour in polar regions is on the one hand highly challenging and on the other hand of high interest for various reasons, among others, due to the highly sensitive response of at least the Arctic to climate change. Thus, the overall topic and objectives of the paper are highly relevant. Retrieval improvements and enhanced applicability are presented and evaluated. Thus, the paper fits into the scope of AMT. However, in my view point the paper requires a few substantial improvements.

From my view perspective the following major points need to be addressed:

1) Section 2.6 introduces one of the new features of the retrieval. Unfortunately references are not properly linked or missing. But even when available it is likely not possible for me to understand the filtering method. I think this method needs to be explained in more detail. I further propose to show spatial maps which display the impact of the mask for one or two days in ,e.g., January and/or July. The objective is to showcase the impact of the screening in product space, other approaches for this are welcome.

It would be very helpful to explicitly mention the conditions (TCWV threshold, surface type) when the retrieval can be applied in, e.g., section 2.5 (i.e., extend the last sentence of section 2.3).

Section 2 introduces a switch between retrievals at 7 kg/m2 and an upper application limit of 15 kg/m2. However, figs 1 and 8 exhibit features at 6 kg/m2 and figs 1 and 3 values above 15 kg/m2. Figure 1 also shows that the majority of values in summer are around and above 15 kg/m2. Please clarify this (seeming) contradiction.

2) It is clear that the availability of ground truth data hampers the evaluation of TWV in the Artic. However, three GRUAN, a few more GUAN and maybe other WMO stations are within the area of interest (and were partly used for retrieval development). I can imagine that data from some stations might exhibit too large values even in winter. Nevertheless, I propose to assess the utilisation of radiosonde data from these sources for evaluation of MHS and AMSU-B over a common period and given sufficient collocations use it in addition to N-ICE data. The joint evaluation of AMSU-B and MHS and the application of the new ice cloud masking using ground-based or in-situ data is currently lacking but would strongly support one of the main objectives of the paper.

3) To me, the current presentation of evaluation results requires rephrasing: I am not a validation expert for the polar regions. However, the interpretation of evaluation results/performances as "good" seems to go too far. I propose to not interpret the results this way and just summarise the results (which might be termed as indicative of successful application to MHS and improvements). Alternatively, a brief summary of existing results from other evaluation efforts in the Arctic can be provided. Given superior quality such statements might be adequate. Depending on results from 1) a successful application to MHS and an improvement via ice cloud masking might have been proven as well.

I don't think that the terminology "benchmark" is adequate for a satellite based TWV product in the Arctic. Please speak of "comparisons" instead.

4) The paper requires careful cross-reading for various reasons. Among them are: partly units are not provided, a few references are not properly linked or missing and various formulations don't seem to be correct. Some of the latter are mentioned below. I am not a native speaker and propose that a native speaker is cross-reading the paper.

In addition I have the following minor comments, partly linked to the comments above:

**) The evaluation results exhibit features caused by changes between retrieval algorithms. A brief discussion on expected impacts of reprocessed products would be adequate. E.g., the application of thresholds can easily lead to temporal and spatial inhomogeneities. I propose that the team can find a more physical solution than the one mentioned in the conclusions to overcome this issue. I recommend to reformulate such potential future plans.**

**) Page 1, line 17: Overall TWV increases due to increases in temperature. I propose to rephrase accordingly.**

**) p 2, l4: Usually a frozen retrieval is applied consistently. However, various other factors are important in this context as well. Please rephrase, i.e., delete "analysis method" and add others, e.g., instrument degradation.**

**) p2, l23: Please delete "successfully" and briefly mention the results (i.e., quality indicators as used in this paper).**

**) p2, l28-29: Please cross-read.**

**) p3, l14, l16, 17: Please delete "will" and remove definition of abbreviation here. Please mention Metop-A and Metop-B explicitly.**

**) Section 2.2: Please mention briefly how the two retrievals are defined (or give reference to Appendix).**

**) Section 2.5: A brief discussion of where - in TWV space - these transitions occur in a climatological sense would be helpful.**

**) Section 2.6: Various references are missing (i.e., appear as "?"). Please provide them.**

**) I propose to change the order of sections 2.5 and 2.4.**

**) p6, l26: "unexpectedly small" – other months have only half of the amount of data. Maybe the feature has other reasons. Please explain.**

**) p6, l29, l30: Please provide unit for RMSD and delete "really".**

**) Section 3, last paragraph: In addition to 3) please mention a systematic high bias plus fairly large outliers.**

**) Would it be better to have two sections: the first paragraph could be section 5 on first discussions towards full spatial coverage? I am not sure if the second paragraph is needed. At least it needs to be rephrased.**

**) p7, l19: It seems to me that "artefacts to be removed" are not discussed subsequently. Please rephrase.**

**) Abstract, conclusions: please note 3).**

**) p8, l11+12: please remove this sentence.**

**) p8, l14: It has not been proven yet if a stable time series can be generated. Please rephrase accordingly.**

**) Fig 6: Are the large areas of values close to 0 kg/m2 realistic? Please adapt color scale such that structures in summer months can be seen.**

**) Fig. 8: Can you explain the feature at ~3.5 kg/m2 in the top right panel? Please define the solid and dashed lines in the caption.**

---

## Referee Comment (RC2) · Anonymous Referee #2 · 23 Oct 2019

This paper expands on previous work of creating a TWV retrieval in the Arctic by adding some improvements and extending from AMSU-B to MHS. I find this paper sufficient for publication after the following comments are addressed.

General Comments

It's unclear to me which platforms of MHS and AMSU-B you are using in this analysis. On Page 3, line 17 you only mention NOAA16, NOAA17, NOAA18, and the Metop (both A and B?) satellites. Are those the only platforms you're using? What about NOAA15 (AMSU-B) and NOAA19 (MHS)? When evaluating your retrieval in Section 3, you mention using an overlap period of 2008-2009, so are you combining all the MHS and AMSU-B platforms together? What about potential differences among the similar sensors? While the MHS sensors are fairly similar to one another, the AMSU-B

sensors have been shown to have very significant calibration differences (see for example, Chung et al, 2013: "Intercalibrating microwave satellite observations for monitoring long-term variations in upper- and midtropospheric water vapor" and Moradi et al, 2018: "Radiometric correction of observations from microwave humidity sounders"). It would be helpful to include a description of where the AMSU-B/MHS data are from and if there are any calibration corrections or intercalibration applied. As shown in the papers I previously listed, NOAA15 and NOAA16 show some significant calibration issues in the 183 GHz channels later in life, so the 2008-2009 time range selected for Section 3 would be impacted, unless you can show that the calibration differences between the sensors do not matter for your retrieval.

Also in regards to the AMSU-B/MHS dataset, it would be nice to see a better description of the data availability and the instrument characteristics. You have the frequencies listed in Table 1, but you don't reference this table until late in the text when it would be helpful to know these details earlier as you are referencing channel numbers. Also, it would be helpful to include a plot or table showing data availability of the instruments. You mention using an overlap year of 2008-2009 in Section 3, but an overlap of which platforms? Without prior knowledge of the sensors this wouldn't make sense. Showing the period of time each sensor was active would help with this.

Specific Comments

Page 6, line 26. "The worst slope... unexpectedly small amount of data". I'm confused about this statement. According to the table, December does not have the lowest number of data points, so it doesn't seem appropriate to say this lowest correlation may be related to the number of data points. And why is the number of data points "unexpected"? It's not clear to me why you say that.

In Figure 8, it appears that you do get a lot of overlap between the AMSRE and AMSU retrievals for July, while in Figure 7 the case that you show has no overlap. It was a little confusing to go from your statement on Page 7, line 26 saying "in summer the overlap

area is zero" and just a few sentences later you show overlap in the summer in Figure 8. Did you apply the ice cloud mask to Figure 7 and that's why there's no overlap? I realize that Figure 7 is just a day and a different year, but it might be better to show a case where there is some overlap just to be more consistent with Figure 8. Also, it appears in Figure 8 that there are double the number of overlap points in the summer as in the winter (left column) which would also seem to contradict Figure 7.

Page 7, lines 22 and 31. In line 22 you say that the upper limit is 15 kg/mˆ2, but then in line 31 you say the upper limit is 7 kg/mˆ2. Which one is it?

Technical Corrections

Define AMSU-B and MHS the first time they are used.

Please include the references currently marked with a "?" on Page 6, line 10 and Page 7, line 10.

Page 6, line 15. Do you mean +/-7 instead of +/-1? Channel 20 is 183+/- 7 GHz.

Page 7, line 25. "red area" - should say "orange".

———————————————————

---

## Author Comment (AC1) · 21 Nov 2019

Comments from the author provided in blue.

The retrieval of water vapour in polar regions is on the one hand highly challenging and on the other hand of high interest for various reasons, among others, due to the highly sensitive response of at least the Arctic to climate change. Thus, the overall topic and objectives of the paper are highly relevant. Retrieval improvements and enhanced applicability are presented and evaluated. Thus, the paper fits into the scope of AMT. However, in my view point the paper requires a few substantial improvements.

From my view perspective the following major points need to be addressed:

1) Section 2.6 introduces one of the new features of the retrieval. Unfortunately references are not properly linked or missing. But even when available it is likely not possible for me to understand the filtering method. I think this method needs to be explained in more detail. I further propose to show spatial maps which display the impact of the mask for one or two days in ,e.g., January and/or July. The objective is to showcase the impact of the screening in product space, other approaches for this are welcome.

Now Section 2.7: Added Figure 12 in Section 4 (Evaluation of changes/improvements in the retrieval) that shows the impacts of the mask for some selected areas through evenly spaced days - each three months approximately - through 2008.

It would be very helpful to explicitly mention the conditions (TCWV threshold, surface type) when the retrieval can be applied in, e.g., section 2.5 (i.e., extend the last sentence of section 2.3).

Section 2.5: Reformulated section 2.5 to better describe the different retrieval regimes and their working conditions, added Table 3 (P30) as summary.

Section 2 introduces a switch between retrievals at 7 kg/m2 and an upper application limit of 15 kg/m2. However, figs 1 and 8 exhibit features at 6 kg/m2 and figs 1 and 3 values above 15 kg/m2. Figure 1 also shows that the majority of values in summer are around and above 15 kg/m2. Please clarify this (seeming) contradiction.

Clarified in Section 2.5 that the switch between retrievals is done in the brightness temperature space, which doesn't correspond to a specific TWV value. We provide approximate limits and switch, but they are not 'hard limits', hence the values above 15kg/m2 in Figures 1 and 13.

2) It is clear that the availability of ground truth data hampers the evaluation of TWV in the Arctic. However, three GRUAN, a few more GUAN and maybe other WMO stations are within the area of interest (and were partly used for retrieval development). I can imagine that data from some stations might exhibit too large values even in winter. Nevertheless, I propose to assess the utilisation of radiosonde data from these sources for evaluation of MHS and AMSU-B over a common period and given sufficient collocations use it in addition to N-ICE data. The joint evaluation of AMSU-B and MHS and the application of the new ice cloud masking using ground-based or in-situ data is currently lacking but would strongly support one of the main objectives of the paper.

Added new subsection on paper (3.2) with a comparison of AMSU-B and MHS derived TWV with GPS and radiosonde data during the period 2008-2009.

3) To me, the current presentation of evaluation results requires rephrasing: I am not a validation expert for the polar regions. However, the interpretation of evaluation results/performances as "good" seems to go too far. I propose to not interpret the results this way and just summarise the results (which might be termed as indicative of successful application to MHS and improvements).

P8 L5-7: Changed most qualifiers from description to fit this comment, and removed 9 outliers from MHS TWV comparison with N-ICE TWV, which improves the performance significantly.

Alternatively, a brief summary of existing results from other evaluation efforts in the Arctic can be provided. Given superior quality such statements might be adequate.

P2 L26-30: Provided the following brief summary from the evaluation papers mentioned previously: In Rinke et al 2009, a comparison with the HIRHAM model showed realistic patterns and maximum root-mean-square

differences for monthly data in summer of 1-2.5 kg/m2. For the comparison with Ny Alesund radiosondes in Palm et al. 2010, the correlation coefficient was 0.86 and the slope 0.8 \pm 0.04 kg/m2. And lastly, in Buehler et al. 2012, ASMU-B TWV are compared to GPS data from Kiruna, with standard deviations of 1kg/m2 and a correlation coefficient of 0.86.

Depending on results from 1) a successful application to MHS and an improvement via ice cloud masking might have been proven as well.

While Figure 13 deals with AMSU-B data masking, the same mask can be applied successfully to MHS, as shown in the already masked maps from Figure 11.

I don't think that the terminology "benchmark" is adequate for a satellite based TWV product in the Arctic. Please speak of "comparisons" instead.

P9,L2 Description of AMSR-E in section 4; P10,l8 Conclusions — > Both mentions rephrased

4) The paper requires careful cross-reading for various reasons. Among them are: partly units are not provided, a few references are not properly linked or missing and various formulations don't seem to be correct. Some of the latter are mentioned below. I am not a native speaker and propose that a native speaker is cross-reading the paper.

Carefully proofread paper.

*In addition I have the following minor comments, partly linked to the comments above:*

**) The evaluation results exhibit features caused by changes between retrieval algorithms. A brief discussion on expected impacts of reprocessed products would be adequate. E.g., the application of thresholds can easily lead to temporal and spatial inhomogeneities. I propose that the team can find a more physical solution than the one mentioned in the conclusions to overcome this issue. I recommend to reformulate such potential future plans.**

We think there is no easy "physical" solution for the discontinuities at the boundaries between the different sub-algorithms. The sub-algorithms, just like many retrieval algorithms that do not use inverse methods, are based on regression analyses which result in "calibration parameters". Thus, in the end, the average state of all atmospheric parameters except water vapour (e.g., the temperature profile) is contained in the calibration parameters.

It is known that the error of the retrieval for each subalgorithm increases strongly when approaching its upper (saturation) limit (see, Melsheimer & Heygster, 2008, Appendix IV) -- Therefore, an appropriate weighted average of the retrieval results of two sub-algorithms in their overlap range (one algorithm nearing its upper limit with increasing errors, the next one still being in it low range with small error) appears meaningful to us.

**) Page 1, line 17: Overall TWV increases due to increases in temperature. I propose to rephrase Accordingly.**

P1L18 Added mention to TWV increase due to temperature

**) p 2, l4: Usually a frozen retrieval is applied consistently. However, various other factors are important in this context as well. Please rephrase, i.e., delete "analysis method" and add others, e.g., instrument degradation.**

P2L8 Rephrased as suggested

**) p2, l23: Please delete "successfully" and briefly mention the results (i.e., quality indicators as used in this paper).**

P2, L27-31 Added short summary of results from the three papers mentioned

**) p2, l28-29: Please cross-read.**

P3, L1-4 Reformulated, ordered sections, added description of new analysis with radiosondes

**) p3, l14, l16, 17: Please delete "will" and remove definition of abbreviation here. Please mention Metop-A and Metop-B explicitly.**

P3, L33 Solved

\#) Section 2.2: Please mention briefly how the two retrievals are defined (or give reference to Appendix).

It is not clear to us what is meant by the "defining" the "two retrievals" - there is the retrieval according to the cited work by Miao (2001), briefly described in section 2.3 (formerly 2.2) and the extended version according to the cited work by Melsheimer and Heygster (2008), briefly described in section 2.4 (formerly 2.3). The calibration parameters used by the different algorithms are discussed in detail section 2.6 (formerly 2.5) and in the appendix. We feel this and the cited references sufficiently define or describe the algorithms and sub-algorithms.

\#) Section 2.5: A brief discussion of where - in TWV space - these transitions occur in a climatological sense would be helpful.

Clarified in Section 2.5 that the switch between retrievals is done in the brightness temperature space, which doesn't correspond to a specific TWV value. We provide approximate limits and switch, but they are not 'hard limits', hence the values above 15kg/m2 in Figures 1 and 13.

\#) Section 2.6: Various references are missing (i.e., appear as "?"). Please provide them.

Section 2.7: References provided (Gonzalez and Woods, 2007), van der Walt et al. (2014))

\#) I propose to change the order of sections 2.5 and 2.4.

Changed order of Sections: Former Section 2.4 is now 2.6, due to changed number of sections.

\#) p6, l26: "unexpectedly small" – other months have only half of the amount of data. Maybe the feature has other reasons. Please explain.

P7L12-13 We wanted to refer to the fact that, for a winter month, this amount of data is small (comparing the 7723324 points in December with the 10691385 in January or 9858305 in February). Clarified this in the text

\#) p6, l29, l30: Please provide unit for RMSD and delete "really".

P7, L15-17. Solved

\#) Section 3, last paragraph: In addition to 3) please mention a systematic high bias plus fairly large outliers.

P8, L15-17. This corresponds to the middle of Section 3.2 now. Mentioned high bias, and large outliers on the context of outlier removal.

---

## Author Comment (AC2) · 21 Nov 2019

Comments from author are in blue
General Comments

P3L6-14: Added Section 2.1 (Data sources) to address the general comments as a whole.

It's unclear to me which platforms of MHS and AMSU-B you are using in this analysis.On Page 3, line 17 you only mention NOAA16, NOAA17, NOAA18, and the Metop(both A and B?) satellites. Are those the only platforms you're using? What aboutNOAA15 (AMSU-B) and NOAA19 (MHS)?

P3L8-9. The description of the possible data sources should have included all satellites with AMSU-B or MHS on board. Fixed that now.

When evaluating your retrieval in Section3, you mention using an overlap period of 2008-2009, so are you combining all theMHS and AMSU-B platforms together? What about potential differences among the similar sensors? While the MHS sensors are fairly similar to one another, the AMSU-B sensors have been shown to have very significant calibration differences (see for exam-ple, Chung et al, 2013: "Intercalibrating microwave satellite observations for monitoring long-term variations in upper- and mid tropospheric water vapor" and Moradi et al, 2018: "Radiometric correction of observations from microwave humidity sounders"). It would be helpful to include a description of where the AMSU-B/MHS data are from and if there are any calibration corrections or intercalibration applied. As shown in the papers I previously listed, NOAA15 and NOAA16 show some significant calibration issues inthe 183 GHz channels later in life, so the 2008-2009 time range selected for Section 3 would be impacted, unless you can show that the calibration differences between the sensors do not matter for your retrieval.

P3L6-14: Summarized in Section 2.1 which platforms the data comes from: Always NOAA-17 for AMSU-B, from the NOAA Fundamental Climate Data Record. Always NOAA-18 for the MHS case.

Also in regards to the AMSU-B/MHS dataset, it would be nice to see a better description of the data availability and the instrument characteristics. You have the frequencies listed in Table 1, but you don't reference this table until late in the text when it would be helpful to know these details earlier as you are referencing channel numbers.

P3L10: Now Table 1 is referenced in Section 2.1.

Also, it would be helpful to include a plot or table showing data availability of the instruments.You mention using an overlap year of 2008-2009 in Section 3, but an overlap of which platforms? Without prior knowledge of the sensors this wouldn't make sense. Showing the period of time each sensor was active would help with this.

P3L10/P 29. Added Table 2 mentioning platforms and sensors.

Specific Comments

Page 6, line 26. "The worst slope... unexpectedly small amount of data". I'm confused about this statement. According to the table, December does not have the lowest number of data points, so it doesn't seem appropriate to say this lowest correlation may be related to the number of data points. And why is the number of data points"unexpected"? It's not clear to me why you say that.

P7L12-13 For a winter month, this amount of data is small (comparing the 7723324 points in December with the 10691385 in January or 9858305 in February). Hence, the description of the

In Figure 8, it appears that you do get a lot of overlap between the AMSRE and AMSU retrievals for July, while in Figure 7 the case that you show has no overlap. It was a little confusing to go from your statement on Page 7, line 26 saying "in summer the overlap area is zero" and just a few sentences later you show overlap in the summer in Figure8. Did you apply the ice cloud mask to Figure 7 and that's why there's no overlap? I realize that Figure 7 is just a day and a different year, but it might be better to showa case where there is some overlap just to be more consistent with Figure 8. Also, it appears in Figure 8 that there are double the number of overlap points in the summer as in the winter (left column) which would also seem to contradict Figure 7.

We address this with two different things:

P9L10-15 I modified the description of Figure 7 (now Figure 11) and the overlap area in general from just indicating that in this particular day there is none to 'in summer this overlap area is small, and in this particular example, the overlap is zero'.

P9L20-25 First, I extended the calculations for Figure 8 (now Figure 13) to three years in total (2006-2008). Additionally, I fixed an issue with this data: in the former version of the paper, there was some spurious overlapping data from the Extended sub-algorithm for Open Water surfaces developed in Scarlat et al, 2018. Hence, the summer overlap between AMSU-B and AMSR-E presented now in Figure 13 is mostly data that needs to be removed by the ice cloud mask.

Page 7, lines 22 and 31. In line 22 you say that the upper limit is 15 kg/m^2, but then in line 31 you say the upper limit is 7 kg/m^2. Which one is it?

P9L7-8 and P9L19. Both are correct, but the phrasing was really vague. Corrected to specify how the different upper limits mentioned in each case are for different surfaces (since the extended algorithm with emissivity information is only applied to sea ice surfaces).

P5L20-31/P30 Reformulated Section 2.5 to better describe the different retrieval regimes and their working conditions, added Table 3 as summary.

Technical Corrections

Define AMSU-B and MHS the first time they are used.

P1L7. Defined in the abstract now

Please include the references currently marked with a "?" on Page 6, line 10 and Page7, line 10.

P6L25; P7L25. Fixed issue

Page 6, line 15. Do you mean +/-7 instead of +/-1? Channel 20 is 183+/- 7 GHz.

P7L1. Yes, I do. Corrected.

Page 7, line 25. "red area" - should say "orange"

P9L11. Corrected.

---

## Referee Report (RR1)

Review of 'Improved water vapour retrieval from AMSU-B/MHS in polar regions' by Triana-Gomez et al.

The authors present an update to a water vapour retrieval suitable for the polar regions, expanded to a more modern sensor (MHS) and now including a screening procedure intended to mitigate the deleterious effects of significant scattering from clouds on the retrieval. The paper presents itself as describing these two 'advances' to the previous retrieval, 'intended as groundwork' for a future planned combined product that would incorporate oceanic microwave imager retrievals as well, combining the two into a pan-Arctic product that could potentially cover a long time period. While the work is nicely presented and well written, I do not believe that this rises to the level of significance to the community that merits publication in this journal. For that reason, I recommend rejection of the manuscript as it stands, with encouraged resubmission if the authors follow through on the stated future work. More in-depth comments follow, broken up into a few major bullet points and minor comments.

1. The abstract lays out the paper as presenting two advances to an old and established retrieval of Arctic water vapour. The first is simply expanding the old retrieval to a new sensor, which is almost a trivial exercise since the channels are almost identical and the difference of absorption characteristics between the 150 vs. 157GHz channels is fairly trivial. It is nice to see that the retrieval works similarly for MHS as it does AMSU-B, and this is well laid out by the authors, but it is not surprising or noteworthy for the community that reads AMT -- perhaps a small technical challenge but not scientifically significant. The abstract's second advance touted is a new screening for artefacts caused by convective clouds. While this is potentially quite interesting, it is essentially a footnote in Section 2 of the manuscript, and the description of this new filtering method is literally restricted to 5 lines of the total manuscript (P6 L23-27). Furthermore, I am not convinced that it is necessarily filtering out 'high cloud ice content in convective clouds' as the abstract states; rather the authors infer that the low retrieved TWV is indicative of a scattering signal from cloud ice, but this is not demonstrated in the paper and thus it appears that it is just an assumption. It could be justified as such if compared to other satellite imagery or an IWP product. The remainder of the paper holds no strong conclusions: 'The improved retrieval performs better when compared to another satellite product and to in situ data' (P9 L32) and 'the results are satisfactory' (P10 L5). It is unclear what exactly is demonstrably better as much of the discussion is qualitative, or even that the updated retrieval could outperform reanalysis datasets, which is an almost necessary test for retrievals to demonstrate.

2. The second major comment has more to do with the methodology upon which the study rests. This is a subjective opinion, but so-called 'non-physical' retrievals such as Miao et al. (2001) were quite important twenty years ago when radiative transfer codes were slower and less advanced, but are becoming less useful today. The results show the downsides to such a regression-based bin method, with big gaps visible in Fig. 1. Why would the community use such a product when reanalyses have no such gaps in coverage or artefacts between bins, not to mention blended TWV products that exist

too? Modeling of sea ice emissivity is of course still a big challenge, but physically-based retrievals from microwave radiometers already exist over sea ice and indeed all surfaces (for example, see the MIRS retrieval from NOAA: https://www.star.nesdis.noaa.gov/mirs/geonwp.php). If regression-based retrievals such as the one presented are to remain relevant, they need to demonstrate their worth relative to similar products, including reanalyses (see e.g. Duncan and Kummerow 2016). If this paper had presented the validation against in-situ sources alongside comparison with say ERA5 data and shown that it outperforms the reanalysis, then it is of much more interest to the community. Even the proposed combined TWV product of this retrieval with RSS data (P10 L13) would need to prove this, and it is as of yet far from certain; the bin-based artefacts are a major concern and merging with RSS would be difficult in itself due to their own biases and simplifying assumptions made for radiative transfer. If the methodology can be shown to outperform physically-based retrievals (with a full forward model) then it has interest for the community, but otherwise it strikes me as requiring corrections on top of corrections that do not lead to greater physical understanding, and could be perhaps be better accomplished by a neural net retrieval.

3. The radiative transfer equation upon which the methodology rests struck me as maybe being incorrect (Eq. 1). If we take the case of surface emissivity of 1, then TB is directly proportional to surface temperature; if we take a fully opaque atmosphere with negligible transmittance (tau>>1), then again the second term goes to zero and TB is again directly proportional to Ts; if surface emissivity were zero, then TB is essentially Ts minus an atmospheric contribution? I apologise if I am misinterpreting this, but it makes no sense to me when I consider these cases. However, it is indeed the exact same equation given in Miao et al. (2001) and originally in Guissard and Sobieski (1994), so I am perplexed. I did not have the time to follow the full derivation in the G&S 1994 paper, but it seems suspect to me. I would suggest examining this in detail to make sure this isn't a typo, because it appears like a form given in Grody (1976) but with Ts and To flipped. Again, apologies if I have misinterpreted this--it just struck me as odd.

Minor comments:
P1 L12: The title uses 'polar' but the paper almost exclusively uses 'Arctic' only. Unless there is some focus on the Southern Hemisphere too the title should be reconsidered.
P2 L1: Is 1m squared a typo?
P2 L10: Fix citation Bobylev and Mitnik
P2 L16: According to OSCAR SSM/T2 confusingly stands for Special Sensor Microwave Humidity (https://www.wmo-sat.info/oscar/instruments/view/535)
P2 L20-21: Is there proof of this statement? A citation or elaboration would be good here.
P3 L10: Is a table with launch dates necessary? It does not really impact the paper.
P3 L16: Typo in citation, Sobieski
P4 L15: What are the units on k? Since absorption coefficients for water vapour are very well known, the derived regression parameters C could be compared against values in the literature.

P5 L17: Perhaps I missed this, but does the manuscript state how the 'surface types are obtained'? This is a key part of the algorithm and surely any future combined product. There is something at P7 L25, but it is unclear if this is how the algorithm functions or if that was just for that particular analysis.

Section 3.1: How are coincident points defined?

P7 L9: Is there any justification for saying that time differences are 'likely' the cause of differences, or is this speculation?

P7 L12: What was the 'expected amount of data'? I found this confusing.

P7 L19 It would be interesting to investigate why there is this 'low agreement in summer' rather than just to 'presume' -- this could possibly be tested by contrasting open water with retrievals over ice.

P8 L5: I don't understand this -- you eliminated the outliers from the analysis and then found that there was good agreement? What was the justification for eliminating the outliers?

P8 L30: The bias values should be smaller than RMSD by definition.

P13 L5: Typo 'Anctarctica'

Fig. 10: I really like the colour scale used, but it seems insufficient for the July panels. Suggest using separate colour scales, one for each season so that patterns over sea ice can be seen in both seasons.

Fig. 12: Some discussion of the third row here seems necessary. Surely it's not physical to expect TWV=14 or more in the southern Hudson bay with TWV<3 just to the south even after screening?

---

## Author Response (AR2)

In **black** we show the original reviewer comments, in **blue** we provide the authors response

**RELEVANT MODIFICATIONS**

We consider that we have fully satisfied the requests of the two reviewers of the first round of reviews and also resolved the majority of concerns of the third, new, reviewer. The only remaining concern is a rather philosophical different opinion on the value of direct satellite retrievals compared to atmospheric reanalysis and integrated retrievals based on radiative transfer models. While we think that this a worthwhile debate for the scientific community it shouldn't hinder the publication of results based on well established and reliable methods accepted by large parts of the community.

The most relevant changes on the article are the following:

- Introduction of a comparison of AMSU-B data to ERA5 reanalysis in new Section 3.3 and a comparison of ERA5 to GPS and radiosondes at the end of Section 3.2. With this, we show that the performance of our retrieval shows similar quality as the ERA5 reanalysis and hence reinforce its usefulness

- Expansion of the Sections 2.7 and 4 regarding the masking of ice cloud artefacts. In Section 2.7 we provide a more detailed description of the method and justification of its need. In Section 4, we provide some additional statistics and expanded/developed further the analysis

- The other changes in the article deal with the minor comments from both reviewers, fixing typos and rephrasing or expanding the relevant sentences (such as the acknowledgements)

**COMMENTS REVIEWER 1**

1) Please acknowledge provision of radiosonde and GPS data and provide information to where you got the data from (e.g., in the acknowledgment).

P12 L17-20: Added the following acknowledgment: The authors would like to thank the Department of Atmospheric Science, University of Wyoming for the radiosonde raw data (http://weather.uwyo.edu/upperair/sounding.html) and the International GNSS Service for the GPS data (ftp://igs.ensg.ign.fr/pub/igs/data/).

2) page 2, line 31: ASMU->AMSU
P2 L28: Corrected

3) p2, l32: I think it should say RMSD and not standard deviation. If so, please change.
P2 L28: Yes, it should. Corrected

4) p3, l11: Eumetsat->EUMETSAT
P3 L6: Corrected

5) p5, l23: wich->which
P5 L23: Corrected

6) p8, l17: Please add „associated with the quasi periodic peaks" after outliers.
P8 L15: Added

7) p8, l22: It is not clear to me what is meant with „homogenized year". Please rephrase.
P8 L21: the word 'year' after homogenized is a typo from previous rephrasing, it has been removed

8) p8, l34: I think this statement is maybe not correct. The bias and the RMSD are small in winter because the values are small. The reason for a low correlation is likely that the temporal coherence is less pronounced.
P9 L9-12: We agree with your statement but do not see how it is in disagreement with what we wrote about that the higher summer RMSD is also seen in Fig. 8. We have reformulated the sentence and added your statement.

9) p9, l6,7: Please remove „The bias…datasets."
P9 L7: Removed

**COMMENTS REVIEWER 2**

The authors present an update to a water vapour retrieval suitable for the polar regions, expanded to a more modern sensor (MHS) and now including a screening procedure intended to mitigate the deleterious effects of significant scattering from clouds on the retrieval. The paper presents itself as describing these two 'advances' to the previous retrieval, 'intended as groundwork' for a future planned combined product that would incorporate oceanic microwave imager retrievals as well, combining the two into a pan-Arctic product that could potentially cover a long time period. While the work is nicely presented and well written, I do not believe that this rises to the level of significance to the community that merits publication in this journal. For that reason, I recommend rejection of the manuscript as it stands, with encouraged resubmission if the authors follow through on the stated future work. More in-depth comments follow, broken up into a few major bullet points and minor comments.

1. The abstract lays out the paper as presenting two advances to an old and established retrieval of Arctic water vapour. The first is simply expanding the old retrieval to a new sensor, which is almost a trivial exercise since the channels are almost identical and the difference of absorption characteristics between the 150 vs. 157GHz channels is fairly trivial. It is nice to see that the retrieval works similarly for MHS as it does AMSU-B, and this is well laid out by the authors, but it is not surprising or noteworthy for the community that reads AMT -- perhaps a small technical challenge but not scientifically significant.

P1 L1-10, Section 3: We agree that it is not surprising that the adaptation of the retrieval to MHS works. As you, however, correctly describe there are differences between the two sensors like the different frequencies and polarization in some channels. We here not only describe the technical part of the adaptation but also prove its success with independent data comparisons. We consider both parts, the adaptation and evaluation, scientifically significant and think that they need to be published so it provides a reference for the extended time series.

The abstract's second advance touted is a new screening for artefacts caused by convective clouds. While this is potentially quite interesting, it is essentially a footnote in Section 2 of the manuscript, and the description of this new filtering method is literally restricted to 5 lines of the total manuscript (P6 L23-27). Furthermore, I am not convinced that it is necessarily filtering out 'high cloud ice content in convective clouds' as the abstract states; rather the authors infer that the low retrieved TWV is indicative of a scattering signal from cloud ice, but this is not demonstrated in the paper and thus it appears that it is just an assumption.It could be justified as such if compared to other satellite imagery or an IWP product.

There is clear evidence that there is an effect of ice clouds at the AMSU-B frequencies as studied in Sreerekha (2005), and has been used related to the low retrieved TWV values in different articles such as (Melsheimer et al., 2016). We have added some additional explanations to justify this need better:

P6 L16-32, Section 2.7: We have expanded the justification of the method

P10 L2-14 Section 4: We have added ERA5 maps to Figure 11 to show visually the problematic regions, i.e., ice clouds, in AMSU-B that are not present in ERA5

The remainder of the paper holds no strong conclusions: 'The improved retrieval performs better when compared to another satellite product and to in situ data' (P9 L32) and 'the results are satisfactory' (P10 L5). It is unclear what exactly is demonstrably better as much of the discussion is qualitative, or even that the updated retrieval could outperform reanalysis datasets, which is an almost necessary test for retrievals to demonstrate.

P9 L13-20, Figure 8: Assessed performance of ERA5 compared to radiosondes and GPS

P9 L21-32, Figure 10 and Table 5: Added new section 3.3, with a comparison with ERA5

2. The second major comment has more to do with the methodology upon which the study rests. This is a subjective opinion, but so-called 'non-physical' retrievals such as Miao et al. (2001) were quite important twenty years ago when radiative transfer codes were slower and less advanced, but are becoming less useful today. The results show the downsides to such a regression-based bin method, with big gaps visible in Fig. 1. Why would the community use such a product when reanalyses have no such gaps in coverage or artefacts between bins, not to mention blended TWV products that exist

too? Modeling of sea ice emissivity is of course still a big challenge, but physically-based retrievals from microwave radiometers already exist over sea ice and indeed all surfaces (for example, see the MIRS retrieval from NOAA: https://www.star.nesdis.noaa.gov/mirs/geonwp.php ). If regression-based retrievals such as the one presented are to remain relevant, they need to demonstrate their worth relative to similar products, including reanalyses (see e.g. Duncan and Kummerow 2016). If this paper had presented the validation against in-situ sources alongside comparison with say ERA5 data and shown that it outperforms the reanalysis, then it is of much more interest to the community. Even the proposed combined TWV product of this retrieval with RSS data (P10 L13) would need to prove this, and it is as of yet far from certain; the bin-based artefacts are a major concern and merging with RSS would be difficult in itself due to their own biases and simplifying assumptions made for radiative transfer. If the methodology can be shown to outperform physically-based retrievals (with a full forward model) then it has interest for the community, but otherwise it strikes me as requiring corrections on top of corrections that do not lead to greater physical understanding, and could be perhaps be better accomplished by a neural net retrieval.

We believe that direct retrievals like the one presented here are still relevant for the scientific community and operational centers, which is also demonstrated by the use of such data in recent publications (using AMSR-E TWV over ocean or the AMSU-B TWV retrieval that we deal with in this paper summarized below). As the reviewer mentions this point is more a "subjective opinion" and should not dominate the assessment of this manuscript. Our retrieval is based on physical principles, which due to the complexity of the sea ice - atmosphere microwave emission need simplifications based on empirical factors. While RTM based retrievals are more complex they also use approximations and simplifications. A descrimination in physical and nonphysical retrievals is not correct. The reviewer already mentions one advantage of such direct retrievals: they are fast. Another advantage is that they do not need any auxiliary data. Thus they are independent and completely based on the satellite measurements.

*Recent publications using AMSR derived TWV (http://www.remss.com/missions/amsr/):*
Casadio, S.; Castelli, E.; Papandrea, E.; et al. (2016): Total column water vapour from along track scanning radiometer series using thermal infrared dual view ocean cloud free measurements: The advanced infra-red water vapour estimator (airwave) algorithm. Remote Sensing of Environment, 172, 1-14. doi:10.1016/j.rse.2015.10.037.
Grossi, M.; Valks, P.; Loyola, D.; et al. (2015): Total column water vapour measurements from gome-2 metop-a and metop-b. Atmospheric Measurement Techniques, 8, 1111-1133. doi:10.5194/amt-8-1111-2015.

*Recent publications using AMU-B derived TWV:*
S. A. Buehler, S. Östman, C. Melsheimer, G. Holl, S. Eliasson, V. O. John, T. Blumenstock, F. Hase, G. Elgered, U. Raffalski, T. Nasuno, M. Satoh, M. Milz, and J. Mendrok. A multi-instrument comparison of integrated water vapour measurements at a high latitude site. Atmospheric Chemistry and Physics, 12(22):10925–10943, 2012. (doi:10.5194/acp-12-10925-2012)
M. Palm, C. Melsheimer, S. Noël, J. Notholt, J. Burrows, and O. Schrems. Integrated water vapor above Ny Ålesund, Spitsbergen: a multisensor intercomparison. Atmospheric Chemistry and Physics, 10(3):1215–1226, 2010.
A. Rinke, C. Melsheimer, K. Dethloff, and G. Heygster. Arctic total water vapor: Comparison of regional climate simulations with observations and simulated decadal trends. Journal of Hydrometeorology, 10(1):113–129, 2009. (doi:10.1175/2008JHM970.1)

We agree with the assessment that such direct retrievals need to show that they are useful and can compete with model based datasets like the ERA5 reanalysis. Therefore we extended the evaluation of our TWV dataset by a comparison with ERA5:
P9 L13-20, Figure 8: Assessed performance of ERA5 compared to radiosondes and GPS
P9 L21-32, Figure 10 and Table 5: Added new section 3.3, with a comparison with ERA5
However, the ERA5 assimilates all radiosonde data and thus no independent evaluation of ERA5 is possible. It is therefore not surprising that ERA5 and the radiosonde agree well. As the GPS TWV

measurements are taken at the same locations as the radiosonde launches, also the agreement to GPS is good. While our direct retrieval does not outperform the ERA5 dataset it does show similar quality. As for our dataset the evaluation is completely independent from the in-situ data and thus meaningful, we can assume that a similar performance is also achieved at other geographical locations. This conclusion cannot be drawn for the ERA5 dataset because of its inherent dependence on the radiosondes measurements. This actually demonstrates another advantage of direct, independent retrievals like the one presented here.

3. The radiative transfer equation upon which the methodology rests struck me as maybe being incorrect (Eq. 1). If we take the case of surface emissivity of 1, then TB is directly proportional to surface temperature; if we take a fully opaque atmosphere with negligible transmittance (tau>>1), then again the second term goes to zero and TB is again directly proportional to Ts; if surface emissivity were zero, then TB is essentially Ts minus an atmospheric contribution? I apologise if I am misinterpreting this, but it makes no sense to me when I consider these cases. However, it is indeed the exact same equation given in Miao et al. (2001) and originally in Guissard and Sobieski (1994), so I am perplexed. I did not have the time to follow the full derivation in the G&S 1994 paper, but it seems suspect to me. I would suggest examining this in detail to make sure this isn't a typo, because it appears like a form given in Grody (1976) but with Ts and To flipped. Again, apologies if I have misinterpreted this--it just struck me as odd.

P3 L21-23, Equation 1: At the end of the authors comment (section of this document titled "Radiative Transfer Equation") a detailed explanation is given about how the equation is correct. We have slightly expanded the description of the term mp to make it less ambiguous. It now reads: "..., and $m_p$ a correction to take into account both a non-isothermal atmosphere and the difference between the surface (skin) temperature, $T_s$, and the temperature of the atmosphere at the ground, $T_0$, ($m_p = 1$ would be the isothermal case and $T_0 = T_s$)""

Minor comments:
P1 L12: The title uses 'polar' but the paper almost exclusively uses 'Arctic' only. Unless there is some focus on the Southern Hemisphere too the title should be reconsidered.
P1L12 Changed it to Arctic

P2 L1: Is 1m squared a typo?
P1L21 No, it is not. Changed the definition to the phrasing "when mentioning atmospheric water content, we refer to the vertically integrated mass in an air column with an area of 1 m^2" to make it more obvious

P2 L10: Fix citation Bobylev and Mitnik
P2L6 Corrected

P2 L16: According to OSCAR SSM/T2 confusingly stands for Special Sensor Microwave Humidity ( https://www.wmo-sat.info/oscar/instruments/view/535 )
P2L12 Changed it

P2 L20-21: Is there proof of this statement? A citation or elaboration would be good here.
P2 L16-18  The problematic statement is "Above this value, two of the 183.31 GHz band channels become saturated and the sensor is not able to "see" through the whole atmospheric column anymore". We cited the relevant work (see below) and added a short sentence elaborating on this: "In other words, when the TWV reaches a certain threshold, the brightness temperature at these AMSU-B channels  does not change with increasing TWV".

The relevant citations are:
- *J. Miao, "Retrieval of atmospheric water vapor content in polar regions using spaceborne microwave radiometry," Alfred-Wegener Inst. Polar Marine Res., Bremerhaven, Germany, 1998.*
- *Melsheimer, C. and Heygster, G.: Improved Retrieval of Total Water Vapor Over Polar Regions from AMSU-B Microwave Radiometer Data, IEEE Trans. Geosci. Rem. Sens., 46, 2307–2322, 2008.* Specifically the following paragraph as a more extended explanation: "we use the term "saturation" in the following sense. When we measure a quantity T in order to determine a parameter W, this of course implies that T does really depend on W, i.e., T = T(W). In reality, this

often applies only to a limited range of W. Typically, when W reaches a certain threshold, T does not change with increasing W anymore. This is generally called saturation. In our case of T being the brightness temperature at one of the AMSU-B channels and W being TWV, T first increases with W, then levels off and starts to decrease slowly (Miao, 1998). This means that at low TWV, the brightness temperature is dominated by the thermal emission of the water vapor and, hence, increases with the water vapor content. Beyond a certain TWV value, the atmosphere becomes opaque, and the brightness temperature comes from the upper part of the atmosphere. The higher the TWV content, the higher—and colder—the portion of the atmosphere that contributes to the brightness temperature; hence, the brightness temperature decreases with increasing water vapor. Beyond a certain TWV value, the atmosphere becomes opaque, and the brightness temperature comes from the upper part of the atmosphere. The higher the TWV content, the higher—and colder—the portion of the atmosphere that contributes to the brightness temperature; hence, the brightness temperature decreases with increasing water vapor content."

P3 L10: Is a table with launch dates necessary? It does not really impact the paper.
P3L8 This was a modification introduced during the last round of revision on the suggestion of a reviewer. Although this table is not required to understand the paper, we feel it provides a more clear explanation about the different coverage periods of the different sensors and satellites. While these dates are available online, it is not straightforward to find them together.

P3 L16: Typo in citation, Sobieski
P3L17 Corrected

P4 L15: What are the units on k? Since absorption coefficients for water vapour are very well known, the derived regression parameters C could be compared against values in the literature
P4L5 Added units on k ($m^2$/kg). Thus, the product of mass absorption coefficient and total water vapour (kg/$m^2$), which is the optical depth, is dimensionless as it should be.. We agree that absorption coefficients are well known. However, the relation between the mass absorption coefficients and the the regression coefficients is complicated and has never been formulated explicitly until now (for each sub-algorithm, the two regression coefficients depend on the three mass absorption coefficients of water vapour at the three frequencies used, see details in *Miao, 1998*, cited above). Thus, even a check if the regression coefficients are consistent with water vapour mass absorption coefficients is beyond the scope and aim of this study.

P5 L17: Perhaps I missed this, but does the manuscript state how the 'surface types are obtained'? This is a key part of the algorithm and surely any future combined product. There is something at P7 L25, but it is unclear if this is how the algorithm functions or if that was just for that particular analysis.
P3 L13-15 Changed the statements in P7 L25 to start of Section 2.1 (P3 L13-15) to reflect that this is how the algorithm functions

Section 3.1: How are coincident points defined?
P7 L13-14, Section 3.1: Now the text explicitly mentions this: "For this analysis, we considered all the coincident points in the daily gridded data with a 0.25º grid."

P7 L9: Is there any justification for saying that time differences are 'likely' the cause of differences, or is this speculation?
P7 L17-19: During the initial analysis, we looked at different parameters that could be the source of these TWV differences, such as the time difference, possible gridding issues and the sub-algorithm associated with each point. We found out that a significant majority of the "high TWV difference" points are associated with time differences. We checked this again, and came to the same conclusion. The word "likely" is probably wrongly used here, since there is an actual justification for this. We changed this sentence in the paper to: "These points are mostly associated with time differences of the satellite overpasses, and amount to only about 0.27% of the data, so they are not significant in the overall picture."

P7 L12: What was the 'expected amount of data'? I found this confusing.
P7 L21: Removed this since it is confusing and not relevant here

P7 L19 It would be interesting to investigate why there is this 'low agreement in summer' rather than just to 'presume' -- this could possibly be tested by contrasting open water with retrievals over ice.
P7 L25-28: We do something along the lines of this suggestion in the "surface study" (Figures 2 and 3) We agree that it would be interesting to go even more in-depth with this, but though we indicate the agreement is lowest it is still very good and it feels unneeded

P8 L5: I don't understand this -- you eliminated the outliers from the analysis and then found that there was good agreement? What was the justification for eliminating the outliers?
P8 L13-14 Rephrased, added elaboration why these outliers should be eliminated and are problematic

P8 L30: The bias values should be smaller than RMSD by definition.
P9 L5-6 Removed sentence.

P13 L5: Typo 'Anctarctica'
P14 L15 Corrected

Fig. 10: I really like the colour scale used, but it seems insufficient for the July panels.
Suggest using separate colour scales, one for each season so that patterns over sea ice can be seen in both seasons.
Fig 10: We consider this a good suggestion, and will implement it for the following versions of the paper.

Fig. 12: Some discussion of the third row here seems necessary. Surely it's not physical to expect TWV=14 or more in the southern Hudson bay with TWV<3 just to the south even after screening?
Fig 12: If we compare ERA5 reanalysis with the AMSUB TWV on that date (you can see the figure below for the date and region in question, 6.08.2008), we can see that the expected - physical and correct - TWV is the one higher or equal to 14kg/m$^2$, both retrievals are in agreement about these values. The remaining couple of pixels in the AMSUB retrieval with ~3.5kg/m$^2$ are just not screened by the mask and are unphysical. We added a short paragraph summarizing this in the analysis of Figure 12, P L "We confirmed by comparison to the ERA5 atmospheric reanalysis that the remaining high TWV values are within the expected range. Also the high, >14 kg/m$^2$, TWV values on 6th July 2008 in the Hudson Bay area are in agreement with ERA5."

[Figure]

**Radiative Transfer Equation**

The reviewer's comment, item 3., seems to show that the brightness temperature seen by the satellite is just proportional to the ground temperature for a ground with emissivity equal to 1, but also for a totally opaque atmosphere (transmittance equal to 0), which would obviously be wrong. The error in the reviewer's argumentation was to assume that the factor $m_p$ would be the same in both cases. Here is a detailed derivation:

The equation in question, i.e., equation (1) in our manuscript, corresponds to equation (27) in Guissard and Sobieski (1994): It describes the brightness temperature at the satellite, above the atmosphere (at $z = H$):

$$T_B(\theta) = m_p T_s - (T_0 - T_c)(1 - \varepsilon_s)e^{-2\tau} \tag{1}$$

where

- $\tau$ is the total optical depth of the atmosphere, i.e. the integral of the atmospheric extinction profile $\alpha(z)$: $\tau = \int_0^H \alpha(z)dz/\cos\theta$; note that Guissard and Sobieski (1994) use $\Upsilon = e^{-\tau}$ instead

- $T_0$ is the atmospheric temperature at the lowest level, $T_0 = T_a(z=0)$ (Guissard and Sobieski (1994) call it $T_a$)

- $T_s$ is the surface (skin) temperature, in general not equal to $T_0$

- $T_c$ is the 3K cosmic background radiation

The whole complexity of radiative transfer is in the factor $m_p$, defined in Guissard and Sobieski (1994), equation (28). It contains two effects: (1) the deviation of the atmosphere from being isothermal, and (2) the fact that $T_s \neq T_0$. The definition is

$$m_p = 1 + \left[ (1 - \varepsilon_s)e^{-\tau}\frac{T_0 - T_s}{T_s} - \frac{I_p}{T_s} \right] \tag{2}$$

where

$$I_p = I_1 + (1 - \varepsilon_s)e^{-2\tau}I_2 \tag{3}$$

and the terms $I_1$ and $I_2$, defined in Guissard and Sobieski (1994), equations (21) and (25), both are vertical integrals related to the atmospheric absorption, weighted by the vertical gradient $T_a'(z)$ of the atmospheric temperature $T_a(z)$, and both vanish for an isothermal atmosphere. It is important to note that $m_p = 1$ holds only if the atmosphere is isothermal *and* $T_0 = T_s$.

Now we set the ground emissivity $\varepsilon_s$ equal to one, so the brightness temperature at the satellite is just

$$T_B(\theta) = m_p T_s \tag{4}$$

However $m_p$ is not equal to one, but still contains several terms related to atmospheric emission and absorption:

$$m_p = 1 + (1 - \mathrm{e}^{-\tau})\frac{T_0 - T_s}{T_s} - \frac{I_1}{T_s} \tag{5}$$

Note that $I_1$ vanishes for an isothermal atmosphere. Inserting this into our equation (4) above, we in fact get, for the brightness temperature at the satellite, for $\varepsilon_s = 1$ and non-opaque atmosphere:

$$T_B(\theta) = m_p T_s = T_s \mathrm{e}^{-\tau} + T_0(1 - \mathrm{e}^{-\tau}) + I_1 \tag{6}$$

The first term describes the emission by the ground, transmitted through the atmosphere, the second term is the upwelling atmospheric radiation for an isothermal atmosphere at temperature $T_0$ and the third term is a correction of the upwelling radiation for a non-isothermal atmosphere.

If, in contrast, we just set $\tau \gg 1$, so we have an opaque atmosphere and $\mathrm{e}^{-\tau} \approx 0$, we also get

$$T_B(\theta) = m_p T_s \tag{7}$$

as above, but the difference lies in $m_p$:

In this case, it is

$$m_p = 1 + \frac{T_0 - T_s}{T_s} - \frac{I_1}{T_s} \tag{8}$$

So we get, for the brightness temperature at the satellite

$$T_B(\theta) = m_p T_s = T_s + (T_0 - T_s) + I_1 = T_0 + I_1 \tag{9}$$

Here, $T_0$ is the upward emission of a totally opaque atmospheric layer of uniform temperature $T_0$, and $I_1$ is then the correction term for the temperature not being uniform.

This should resolve item 3. We have noticed that our brief description of the factor $m_p$ after equation (1) in our manuscript was not complete and have corrected that. We now state:

[revised manuscript text omitted]

---

## Author Response (AR3)

In **black** we show the original reviewer comments, in **blue** we provide the authors response

**RELEVANT MODIFICATIONS**
- Rewrote a paragraph of Section 2.7 adding some clarifications as suggested
- Added paragraph at the start of Section 3.3, according to suggestions from second reviewer

**COMMENTS REVIEWER 1**

Minor updates (page and line refer to track change version):
**Page 3, line 19: Please define ASI. Also use simple present here and rephrase.**
P3 L13 : Rephrased and added ASI definition (ARTIST Sea Ice)

**p3, l29: Remove comma after T_0.**
P3 L23 : Comma removed

**section 2.7: Here a few more details/clarifications are helpful:**
Section 2.7, P6 L16 - P7 L2. With all the changes the paragraph is as follows:
"Therefore, image processing methods that rely on the size of ice cloud artefacts can be used: Our approach for eliminating the affected TWV is to find connected areas -- minimum of two pixels -- of low TWV ($<4$ kg/m$^2$) smaller than 50 pixels which are surrounded by higher or non-retrieved values. The threshold of 50 pixels was selected because -- with the data on the selected latitude-longitude grid of 0.25º -- it would approximate to areas of 7000 km$^2$ at 60ºN and 19600 km$^2$ at 80º N, and it amply covers the scale of events that need masking. Then, we remove these connected areas with a succession of morphology operations (Gonzalez and Woods, 2007), using the tools for Python described in van der Walt et al. (2014): First a dilation with a 7x7 square structural element, and then a closing with the same size structural element. We ensure that only the data within the original connected areas are removed by using an image comparison between the mask and the initial connected areas."
Please provide the approximate size or range of sizes, in km2, of 50 pixels.
P6 L31: 50 pixels are around 7000km$^2$ at 60º and 19600km$^2$ at 80º of latitude. Rephrased sentence
Did you apply dilation and then closing? Or did you apply only closing, i.e., dilation and then erosion (p7, l7)? When I understand the method correctly it can alter the original shape of the connected area. Do you ensure that only data within the original shape will be masked? I assume the answer is yes and if so, please mention this.
P7 L1: Yes, now this is mentioned
Please mention that convective clouds larger than a specific size (please define) are not masked and comment its impact, i.e., recall what has been said already in the new version.
P6 L30-31: Added a sentence regarding this
How is the shape detected, does it play a role during detection? Can you either explain this or remove „and shape" from p6, l34-35.
P6 L27: Removed

**p7, l11: Please consistenly use simple present.**
P7 L4-5: Simple present is used now

**p8, l29: Please remove „the" after „resulting".**
P8 L28: Removed

**p9, l26: What is meant with „underestimate"? Correlation coefficients are not a measure of systematic differences.**
P9 L26: We wanted to use underestimate in this case to refer only to the slope, but we can see how the paragraph indicated this also referred to the correlation coefficient. Clarified the sentences.

**p10, l4: Change „ow" into „Low".**
P9 L34: Changed (parallel to the one-to-one line. Low AMSU-B TWV values compared to high)

**p10, l8: Please change „for the central months of the year" into „for summer months".**
P8 L30: Changed
**p10, l9: Please change: „Minimum is of 1.004 kg/m2 in March" into e.g. „The minimum (1.00 kg/m2) is observed in March". You might consider to adapt table 5 accordingly (e.g., change 1 into**

1.00).

P10 L6: Changed both number in text and in Table 5 (P35)

**Please acknowledge provision of ERA5 data.**

P12 L25: Acknowledged in the following sentence "and Copernicus Climate Change Service for the ERA5 data"

**Figure 11: Remove comma after „6 January":**

P26, Fig 11 caption: Removed

**COMMENTS REVIEWER 2**

Second review of 'Improved water vapour retrieval from AMSU-B/MHS in the Arctic' by Triana-Gomez et al.

The authors are commended on their in-depth and thoughtful responses to the many comments and criticisms given by myself and the other reviewers in the previous round of peer review. My main hesitations about allowing this paper through toward publication in AMT have been allayed. The expansion of the analysis to include comparison with ERA5 makes it a much stronger paper, and its argument overall is much improved. The conclusions now better match the results presented, with the language more careful and exact. I have given below one small issue that I would like to see rectified, plus a few very minor textual comments, but I recommend that the paper be published and I do not require seeing it again. Thanks again to the authors for the excellent response and the improved manuscript.

A note on ERA5 section:

In the response to reviewers, the authors state that ERA5 is not an independent estimate of TWV due to its assimilation of some of the observations used as verification here, namely radiosondes. This is a key point, and it would be best if this were mentioned explicitly in the manuscript. I would suggest adding a paragraph to the beginning of Section 3.3 stating what the ERA5 reanalysis data represents, in that ERA5 assimilates radiosonde data, but also that it is a hybrid of model and observations. It is unfortunate that there is not yet a peer-reviewed overview paper of ERA5 and its satellite data usage specifically, but in this context it is worth mentioning that ERA5 does assimilate some 183GHz data over sea ice and snow-covered surfaces (see Table 1 in Bormann et al. 2017 for data usage in the IFS for a version quite similar to that used in ERA5). Specifically, the higher peaking MHS channels are currently assimilated over such surfaces (see link below for real-time example), and some SSMIS 183GHz (F-17) data have been assimilated over snow and sea ice for several years now. It is unclear from the documentation exactly which sensors would've been available and assimilated within ERA5 in the specific months analysed in this study. However, this may provide useful context for readers, in that ERA5 estimates of TWV are neither independent of radiosondes nor entirely independent of microwave humidity sounder radiances. I don't think this does or should change any of the authors' conclusions, but it is important to note.

P 9 L25-30: Thank you for the suggestion and the provided references. We added the following paragraph to the start of section 3.3: "The reanalysis product ERA5 combines a variety of observations and a numerical model using an optimization procedure. Due to ERA5 assimilation of some of the observations used as verification here, namely radiosondes, it is not a completely independent estimate of TWV.  ERA5  also assimilates some 183GHz data over sea ice and snow-covered surfaces (as suggested in Bormann et al., 2017), including the  MHS sounding channels. While it is unclear to the authors which sensors would have been available and assimilated within ERA5 for the time period 2008-2010 of this study, we cannot presume that ERA5 TWV is entirely independent of microwave humidity sounder radiances."

Bormann et al. paper: *--------
Link for current MHS data usage in the IFS:
https://www.ecmwf.int/en/forecasts/charts/obstat/mhs_allsky_mhs_allsky__geo_0001_plot_o_geo_mhs_allsky_mhs_allsky?facets=Category,Satellite%20Data%3BInstrument,SSMIS&time=2020041800&Satellite=METOP-B&Channel=4&Data=STDV%20Obs&Flag=Used

Minor comments:
P9 L24: Typo or missing text: 'ow'
P9 L34: Missing L, it was supposed to say low
Fig. 8: Change 'AMSU' on y-axis title to 'AMSU-B'
P23, Fig. 8: Changed

[revised manuscript text omitted]